# GAPRUNE: GRADIENT-ALIGNMENT PRUNING FOR DOMAIN-AWARE EMBEDDINGS

**Yixuan Tang , Yi Yang**
The Hong Kong University of Science and Technology
`ytangch@connect.ust.hk, imyiyang@ust.hk`

## ABSTRACT

Domain-specific embedding models have shown promise for applications that require specialized semantic understanding, such as coding agents and financial retrieval systems, often achieving higher performance gains than general models. However, state-of-the-art embedding models are typically based on LLMs, which contain billions of parameters, making deployment challenging in resource-constrained environments. Model compression through pruning offers a promising solution, but existing pruning methods treat all parameters uniformly, failing to distinguish between general semantic representations and domain-specific patterns, leading to suboptimal pruning decisions. Thus, we propose GAPrune, a pruning framework that addresses this challenge by considering both domain importance and preserving general linguistic foundation. Our method uses Fisher Information to measure importance and general-domain gradient alignment to assess parameter behavior, then combines these signals using our Domain Alignment Importance (DAI) scoring. Lower DAI scores indicate that the parameter is either less important for the domain task or creates conflicts between domain and general objectives. Experiments on two domain benchmarks, FinMTEB and ChemTEB, show that GAPrune maintains performance within 2.5% of dense models in one-shot pruning at 50% sparsity, while outperforming all baselines. With retraining in 100 steps, GAPrune achieves +4.51% improvement on FinMTEB and +1.73% on ChemTEB, demonstrating that our pruning strategy not only preserves but enhances domain-specific capabilities. Our findings demonstrate that principled pruning strategies can achieve model compression and enhanced domain specialization, providing the research community with a new approach for development. Our code is publicly available at `https://github.com/yixuantt/GAPrune`.

## 1 INTRODUCTION

The deployment of large language models in specialized domains has revealed a critical challenge: while general-purpose models excel at broad language understanding, they often fail to capture domain-specific semantics crucial for real-world applications (Gu et al., 2021; Yao et al., 2024). This semantic gap is evident for embedding models, where precise representation of domain-specific concepts directly impacts downstream task performance. Consider the financial domain: "liability" inherently carries negative sentiment due to its association with obligations and risks, contrasting with its neutral denotation of legal responsibility in general usage. Balyasny Asset Management, a leading quantitative investment firm, has reported that domain-specific embeddings demonstrate significantly higher sensitivity to such financial concepts compared to general-purpose models (Anderson et al., 2024). Similarly, in biochemistry, understanding that "binding" refers to molecular interactions rather than document binding can be crucial for drug discovery pipelines (Kasmaee et al., 2025). This raises the need for domain adaptation.

Most existing approaches for better domain adaptation follow a straightforward scaling paradigm: fine-tune increasingly larger pre-trained language models to capture domain-specific knowledge with designed training data. For example, BMEmbed (Wei et al., 2025) synthesizes specialized training data for a domain-specific retriever. CodeXEmbed (Liu et al., 2024) develops a series of embedding models for code retrieval ranging from 400M to 7B parameters using code-related data,

with performance consistently improving as model size increases. This trend aligns with established scaling laws that predict better performance with more parameters (Kaplan et al., 2020).

However, this scaling-centric approach creates a deployment paradox in real-world applications: while larger models deliver superior results, computational efficiency may drive real-world adoption. Current usage patterns illustrate this efficiency-performance trade-off. At the time of writing, Qwen3-Embedding compact 0.6B model (Zhang et al., 2025) has garnered 3.37M downloads versus only 382K for the higher-performing 8B variant. This is a nearly 9-fold difference that highlights how computational constraints drive adoption decisions over raw performance. This introduces a critical research question: *How can we perform domain adaptation for embedding models while achieving better deployment efficiency?*

Model compression through pruning offers a promising solution, potentially reducing model size by 30-50% while maintaining acceptable performance (Frantar & Alistarh, 2023; Zhang et al., 2024). Yet existing pruning methods face a mismatch when applied to pruning models for domain adaptation. Traditional approaches, whether using magnitude-based pruning (Han et al., 2015) or layer pruning with retraining (Zhang et al., 2024), evaluate parameter importance through a uniform lens, treating all parameters equally regardless of their role in domain adaptation. This uniform treatment creates two critical failure modes. First, parameters encoding crucial domain-specific knowledge might appear unimportant from a general perspective and be incorrectly removed during pruning (Bhattacharyya & Kim, 2025; Zhang et al., 2024). Conversely, calculating importance scores solely based on domain samples may cause models to lose essential general linguistic capabilities, ultimately degrading overall performance (Williams et al., 2025; Zhang et al., 2024). The result is pruned models that either lose their specialized representation or compromise their fundamental abilities.

To address these limitations, we propose Gradient-Alignment Pruning (GAPrune), a novel framework that explicitly balances domain-specific importance with general representation capabilities. Unlike existing methods that evaluate parameters through a single lens, GAPrune measures each parameter across two critical dimensions: (1) its importance for domain-specific performance, and (2) its alignment between general and domain-specific objectives. Our method leverages Fisher Information (Theis et al., 2018) to quantify parameter importance and introduces cross-domain gradient analysis to assess objective alignment. By combining these signals using our **D**omain **A**lignment **I**mportance (DAI) scoring, GAPrune can identify parameters that are both important in the domain and well-aligned with the general objective. Lower DAI scores indicate that the parameter is either less important for the domain task or creates conflicts between domain and general objectives. Experiments on FinMTEB and ChemTEB domains show that GAPrune maintains performance within 2.5% of dense models in one-shot pruning at 50% sparsity, while outperforming all baselines. With retraining in 100 steps, GAPrune achieves +4.51% improvement on FinMTEB and +1.73% on ChemTEB, demonstrating that our pruning strategy not only preserves but enhances domain-specific capabilities. Our work provides both practical tools for deploying efficient domain-specific models and theoretical insights into the nature of domain knowledge encoding.

## 2 RELATED WORK

Our work builds on three research areas: LLM-based embedding models, domain-specific embedding adaptation, and model compression for embedding models.

**LLM-based Embedding Models.** Modern embedding models have evolved beyond traditional architectures to incorporate instruction-following capabilities. Models such as E5-Mistral-Instruct (Wang et al., 2024) and Qwen3-Embedding (Zhang et al., 2025) can process task-specific instructions like "Given a financial question, retrieve relevant documents" and generate embeddings tailored to the specific task. Recent work has further extended this capability, with models like bge-en-icl Li et al. (2025a) leveraging in-context learning to enhance downstream performance. Unlike earlier approaches that relied on smaller models such as BERT (Reimers & Gurevych, 2019), these instruction-tuned models leverage extensive pre-training on diverse text corpora to achieve multi-task capabilities such as STS and retrieval. However, this flexibility comes at a computational cost, as these models typically contain billions of parameters, making efficient deployment a critical challenge for practical applications.

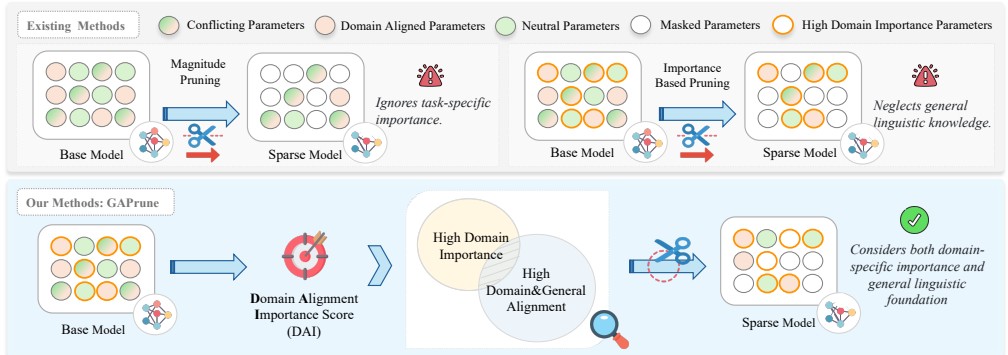

Figure 1: GAPrune framework overview. We compute Fisher Information for domain-specific importance and cross-domain gradient alignment. The Domain Alignment Importance (DAI) score combines these signals to identify parameters that 1) are important for domain performance and 2) are well-aligned for the general and domain-specific objective. Pruning removes parameters with low DAI scores.

**Domain-Specific Embedding Models.** The demand for domain-specific embedding models has grown significantly as there are more applications that require specialized semantic understanding, such as code agents (Li et al., 2024) and high-stakes domains such as finance and healthcare (Anderson et al., 2024; Michalopoulos et al., 2021). Recent benchmarks like CoIR (Li et al., 2025b), FinMTEB (Tang & Yang, 2025) and ChemTEB (Kasmaee et al., 2024) also demonstrate that domain-specific embeddings significantly outperform general-purpose models on specialized tasks. To achieve such specialization, various adaptation training strategies have been developed such as BMEmbed (Wei et al., 2025). However, deploying these large domain-specific embedding models in resource-constrained environments presents significant challenges.

**Model Compression for Embedding Models.** The field of model pruning has progressed significantly from its early magnitude-based foundations (Han et al., 2015). LeCun et al. (1989) pioneered the concept of optimal brain damage, using second-order derivatives to identify less critical parameters, while Li et al. (2017) showed that removing entire computational units through structured pruning could achieve better hardware efficiency than unstructured approaches. Modern large language model pruning methods like SparseGPT (Frantar & Alistarh, 2023) and Wanda (Sun et al., 2024) have pushed the boundaries further, with SparseGPT achieving 50% sparsity through Hessian-based one-shot pruning and Wanda incorporating both weight magnitudes and activation patterns. However, these methods are designed for generative LLMs and face unique challenges when applied to embedding models. Unlike generative models evaluated on perplexity and generation quality, embedding models are assessed using task-specific metrics such as nDCG@10 for retrieval and accuracy for classification (Muennighoff et al., 2023). This difference in evaluation creates unique parameter sensitivity patterns, where embedding models exhibit higher sensitivity to attention head removal as different heads capture distinct semantic relationships essential for representation quality (Voita et al., 2019). Furthermore, existing pruning methods largely treat all parameters uniformly, failing to distinguish between those essential for general linguistic understanding and those specific to particular domains. Recent work by Zhang et al. (2024) and Williams et al. (2025) has demonstrated that domain-aware pruning strategies can better preserve critical domain knowledge, but domain-aware pruning strategies specifically designed for LLM-based embedding models remain underexplored, representing a critical gap in current compression literature.

## 3 GAPRUNE: GRADIENT-ALIGNMENT PRUNING

In this section, we propose GAPrune, a pruning framework that characterizes parameters along two dimensions: their importance for domain-specific performance and their alignment between general and domain-specific objectives. By understanding these patterns, GAPrune makes pruning decisions that preserve domain-specific knowledge while achieving substantial compression.

## 3.1 PROBLEM FORMULATION: DOMAIN PRUNING

Consider a pre-trained embedding model $\mathcal{M}$ with parameters $\theta \in \mathbb{R}^d$. Traditional pruning approaches typically rely on magnitude-based criteria or single-objective importance scores, implicitly assuming that parameter importance is universal across tasks. However, this assumption may break down in domain adaptation scenarios where parameters exhibit domain-dependent behavior: some parameters encode general semantic representations crucial for semantic foundations, while others capture domain-specific patterns.

We formulate domain-specific pruning as a constrained optimization problem that seeks to minimize performance degradation on the target domain while maintaining a desired sparsity level. Let $m \in \{0, 1\}^d$ be a binary pruning mask. Our objective is:

$$\min_{m} \quad \mathcal{L}_{\text{dom}}(\theta \odot m) - \mathcal{L}_{\text{dom}}(\theta) \tag{1}$$

$$\text{s.t.} \quad \|m\|_0 \leq k, \quad k = \lfloor (1-s) \cdot d \rfloor \tag{2}$$

where $s \in [0, 1]$ is the target sparsity ratio, $k$ is the number of parameters to retain, $d$ is the total number of parameters, and $\odot$ denotes element-wise multiplication. The constraint $\|m\|_0 \leq k$ ensures that at most $k$ parameters are retained, achieving the desired compression ratio.

## 3.2 METHOD

GAPrune operates through three sequential stages. First, we sample representative subsets from both general and domain-specific datasets to enable efficient gradient computation. Second, we characterize each parameter using Fisher Information for importance estimation and gradient cosine similarity for cross-domain alignment analysis. Finally, we combine these signals into a unified importance score guided by Information Bottleneck principles, enabling principled trade-offs between compression and domain expertise preservation. Algorithm 1 presents the complete GAPrune procedure.

### 3.2.1 DATA PREPARATION AND SAMPLING STRATEGY

**Data Sources and Format.** Our approach requires two datasets: a general dataset to capture universal linguistic patterns and a domain-specific dataset to capture specialized knowledge. The general dataset contains contrastive triplets constructed from diverse text sources, including news articles, encyclopedic entries, and conversational text, ensuring coverage of different linguistic patterns. The domain-specific dataset contains triplets tailored to the target application, such as financial triplets built from financial reports, or biomedical triplets from clinical notes.

Each data sample is structured as a contrastive triplet $(q, p, n)$, where $q$ is a query text serving as the anchor, $p$ is a positive document semantically similar to the query (discussing the same topic or concept), and $n$ is a negative document semantically dissimilar to the query. This triplet format enables us to compute gradients using InfoNCE Loss (Oord et al., 2018).

**Representative Sampling.** For computational efficiency, we distill the essential statistical properties of both datasets into small, representative subsets that preserve the gradient patterns necessary for parameter analysis. Our sampling strategy employs k-means clustering on the embedding space to select 5,000 representative samples from each dataset while ensuring diverse coverage of the semantic space.

Specifically, we use Qwen3-Embedding-0.6B (Zhang et al., 2025) to generate embeddings for the query $q$ in each triplet. The k-means clustering with $k = 5000$ centroids and 20 iterations. For each centroid, we identify the nearest data point in the embedding space, ensuring that the selected samples are distributed across different semantic regions. This approach guarantees that our representative subset captures the full diversity of linguistic patterns present in the original datasets. These carefully constructed triplets provide the foundation for the following stages. We investigate the impact of calibration data size in Appendix G, demonstrating that GAPrune remains robust even with fewer samples.

### 3.2.2 Characterizing Parameter Behavior

To understand how each parameter contributes to model performance, we analyze it from two complementary perspectives. First, we measure its importance for maintaining performance using Fisher Information, which quantifies how sensitive the model's predictions are to changes in that parameter. Second, we examine how the parameter affects the relationship between general and domain-specific objectives through gradient alignment analysis.

**Importance Estimation via Fisher Information.** A key challenge in domain-specific pruning is determining which parameters are truly essential for downstream performance. We employ Fisher Information (Theis et al., 2018) to quantify parameter importance. Fisher Information measures the expected curvature of the loss landscape around each parameter, providing insight into how much the model's performance would degrade if that parameter were removed or significantly altered. Intuitively, parameters with high Fisher Information are those where small changes lead to large changes in the model's output, making them critical for maintaining performance.

For each parameter $\theta_j$, we approximate the diagonal Fisher Information as:

$$\hat{F}_{jj} = \frac{1}{N} \sum_{i=1}^{N} \left( \frac{\partial \mathcal{L}_i}{\partial \theta_j} \right)^2 \tag{3}$$

where $N$ is the number of calibration data samples and $\mathcal{L}_i$ is the InfoNCE (Oord et al., 2018) loss for triplet $i$. We compute this separately for general data ($F_{jj}^{gen}$) and domain-specific data ($F_{jj}^{dom}$).

**Gradient Alignment for Cross-Domain Analysis.** While Fisher Information reveals parameter importance, it does not capture how parameters interact across different domains. We introduce gradient alignment analysis to understand whether a parameter contributes positively or negatively to both general and domain-specific goals.

Our approach computes gradients using the InfoNCE loss on contrastive triplets from both general and domain-specific datasets. For each parameter $\theta_j$, we calculate gradients $g_j^{gen}$ and $g_j^{dom}$ by accumulating gradients over multiple batches and computing their average. Specifically, we backpropagate through the InfoNCE loss (Oord et al., 2018) computed on batches from the general and domain-specific datasets, then average the resulting gradients across all processed batches. This averaging is used for obtaining stable and representative gradient estimates. Individual batch gradients can be noisy and may not accurately reflect the true gradient direction for the entire dataset. By averaging gradients across multiple batches, we obtain more robust estimates that better represent the overall optimization landscape for each domain.

We then measure gradient alignment using cosine similarity between these averaged gradient vectors:

$$s_g^j = \frac{\langle g_j^{gen}, g_j^{dom} \rangle}{\|g_j^{gen}\| \|g_j^{dom}\| + \varepsilon} \tag{4}$$

where $g_j^{gen}$ and $g_j^{dom}$ represent averaged gradients computed on general and domain-specific data, respectively, and $\varepsilon$ prevents division by zero.

The alignment score $s_g^j \in [-1, 1]$ reveals a valuable insight about parameter behavior in domain-adaptive pruning. When $s_g^j > 0$, the parameter exhibits consistent behavior across domains, suggesting it encodes shared knowledge that benefits both general linguistic understanding and domain-specific tasks. Such parameters represent the core semantic foundations that should be preserved to maintain model versatility. When $s_g^j \approx 0$, the parameter serves distinct roles in different contexts, indicating specialized functionality that requires careful evaluation of its domain-specific importance before pruning decisions. We empirically validate this domain-dependent parameter behavior in Appendix K, showing that parameter importance rankings shift significantly across domains. When $s_g^j < 0$, the parameter demonstrates conflicting contributions to general and domain objectives, suggesting that this parameter contains knowledge that is beneficial for one domain but potentially harmful for the other, making it a candidate for removal when prioritizing domain-specific performance.

### 3.2.3 DOMAIN-AWARE PRUNING STRATEGY

The core insight of our pruning strategy is to identify parameters that are both important for domain-specific performance and well-aligned with general linguistic objectives, while removing those that create conflicts between these two goals. This dual-criteria approach ensures that the pruned model maintains both specialized domain representation and general semantic capabilities.

With each parameter characterized by its domain-specific importance and its cross-domain alignment, we now introduce a pruning strategy that synthesizes these signals. Our approach is guided by the Information Bottleneck (IB) principle (Tishby et al., 2000), which posits that an optimal representation should be maximally informative about a target variable while being minimally complex with respect to the original input. In our context, this translates to finding a parameter sub-network that maximizes fidelity to the domain-specific task while dropping information that creates conflicts between general and domain-specific objectives.

To operationalize this principle, we formulate the **D**omain-**A**lignment **I**mportance (DAI) score, which directly embodies the IB trade-off. The score evaluates each parameter $\theta_j$ by balancing its utility for the domain task against the representational cost of retaining its general-domain knowledge, while considering the nature of its interaction and parameter magnitude:

$$\text{DAI}_j = \left( (F_{jj}^{\text{dom}} - \beta \cdot F_{jj}^{\text{gen}}) \cdot |\theta_j| + \gamma \cdot \sqrt{|\theta_j|} \right) \cdot (1 + \alpha \cdot s_g^j) \tag{5}$$

The first term, $(F_{jj}^{\text{dom}} - \beta \cdot F_{jj}^{\text{gen}}) \cdot |\theta_j|$, forms the core of our IB-inspired importance metric. It prioritizes parameters with high domain-specific Fisher Information ($F_{jj}^{\text{dom}}$) while penalizing those that are primarily important for the general domain ($F_{jj}^{\text{gen}}$), weighted by their magnitude $|\theta_j|$. The hyperparameter $\beta \geq 0$ controls the strength of this penalty, thus steering the trade-off between domain specialization and the preservation of general capabilities. This term effectively quantifies the "net value" of a parameter for the target domain, scaled by its magnitude to reflect both importance and representational capacity.

The second term, $\gamma \cdot \sqrt{|\theta_j|}$, provides a magnitude-based regularization that encourages the retention of parameters with substantial representational capacity, even when their Fisher Information scores are moderate. This helps maintain model expressiveness while still allowing for domain-specific optimization.

The third term, $(1 + \alpha \cdot s_g^j)$, serves as a crucial modulator that refines the score based on cross-domain gradient alignment. It rewards parameters that serve multiple objectives; when a parameter contributes to both domains ($s_g^j > 0$), its importance is increased, as it encodes knowledge beneficial for both general understanding and the specific task. Conversely, when a parameter's influence conflicts with the domain objective ($s_g^j < 0$), its score is reduced. This allows the model to selectively prune parameters that introduce counterproductive interference, thereby resolving optimization conflicts. The hyperparameter $\alpha > 0$ dictates the sensitivity to this alignment signal. In our experiments, $\beta$ is set to 1.0, $\alpha$ is set to 0.2, and $\gamma$ is set to 0.5, providing a balanced influence of importance signals, magnitude information, and alignment information. We provide a detailed ablation study on the contribution of each DAI component in Appendix F and a sensitivity analysis of these hyperparameters in Appendix H.

We apply a one-shot pruning mask by retaining the top-$k$ parameters with the highest DAI scores. This approach creates a compressed model that is both sparse and domain-specialized, keeping parameters that work well for both general and domain-specific tasks while removing those that create conflicts. We provide detailed computational efficiency analysis in Appendix E.

## 4 EXPERIMENT

In this section, we test our GAPrune approach across different settings to demonstrate its performance in domain-specific compression.

## 4.1 EXPERIMENTAL SETUP

**Domains and Datasets.** We test our method on two domains: finance and chemical. These domains present different challenges for embedding model compression and require domain-specific representations to achieve good performance. The finance domain contains specialized terminology like "liquidity ratios" and "market capitalization" that require domain expertise, along with regulatory language and quantitative expressions uncommon in general text. The chemical domain presents highly technical vocabulary with systematic nomenclature, molecular formulas, and complex entity relationships that create semantic patterns distinct from general language.

For computing DAI scores as described in our method, we use three datasets. Our general dataset consists of contrastive triplets from Tang & Yang (2024), sampled from publicly available datasets such as MSMARCO (Bajaj et al., 2016) and SQuAD (Rajpurkar et al., 2016). This provides broad coverage of semantic relationships across diverse text types. For the finance domain, we use the synthesized embedding training dataset from (Tang & Yang, 2025). For the chemical domain, we construct our dataset from the chemistry subset of peS2o (Soldaini & Lo, 2023). We use GPT-4o-mini (OpenAI, 2025) to generate queries based on the corpus, then use these queries as anchors, the original corpus as positive documents, and employ hard negative sampling with all-MiniLM-L6-v2 (Reimers & Gurevych, 2019) to generate negative documents.

All datasets follow the contrastive triplet format $(q, p, n)$ where $q$ is a query, $p$ is a positive document, and $n$ is a negative document. We sample 5,000 examples from each dataset and ensure no overlap with the evaluation set.

**Models and Compression Ratio.** Our experiments use two embedding models with different architectures: Qwen3-Embedding-4B (Zhang et al., 2025) and e5-mistral-instruct (Wang et al., 2024). These models represent recent LLM-based multi-task embedding models with instruction-following capabilities. We test two compression ratios: 30% and 50% sparsity on the MLP layers to examine the trade-off between compression and performance preservation.

**Baselines.** We compare GAPrune against several pruning baselines:

- **Dense:** Corresponding dense model without pruning.

- **Random Pruning:** Randomly selects parameters for removal.

- **Magnitude-based Pruning (Han et al., 2015):** Remove parameters with the smallest absolute values, assuming that weights with lower magnitudes contribute less to model performance. The method calculates importance scores as $|w|$ and prunes parameters below a threshold.

- **Fisher Pruning (Theis et al., 2018):** Uses Fisher Information weighted by parameter magnitude for importance scoring. We test two variants: Domain Fisher Pruning computed from domain datasets and General Fisher Pruning computed from general datasets.

- **$L^3$ Prune (Thennal et al., 2025):** A layer-wise LLM-based embedding model pruning method based on the model's initial loss that removes entire layers. Following the original paper, we only compare this method in the re-training evaluation. We use the small variant of $L^3$ Prune as it prunes similar parameters as our method.

**Evaluation Benchmarks.** The evaluation benchmarks are FinMTEB (Tang & Yang, 2025) and ChemTEB (Kasmaee et al., 2024). For FinMTEB, we evaluate 8 classification, 2 semantic textual similarity (STS), and 8 retrieval tasks. We select these subtasks because they are the most challenging and representative of domain-specific capabilities. ChemTEB contains 17 classification tasks and 2 retrieval tasks. The detailed evaluation instructions are provided in the Appendix B. All the scores reported in experiments are the main score for each task and are detailed in the Appendix C

**Evaluation Protocol.** We conduct two types of evaluation: One-shot Pruning Evaluation and Prune-and-Retrain Evaluation. In one-shot pruning, we apply the pruning mask directly to the pre-trained model and evaluate performance without any additional training. This setting tests whether our DAI scoring can identify parameters that maintain performance immediately after pruning. In the prune-and-retrain evaluation, we first apply the pruning mask, then retrain the pruned model using the corresponding domain dataset with InfoNCE loss (Oord et al., 2018) for 100 steps. This setting

examines whether our pruning strategy provides a good foundation for post-pruning optimization and domain adaptation.

## 4.2 ONE-SHOT PRUNING EVALUATION

Table 1: One-shot Pruning Results on FinMTEB and ChemTEB Datasets. The Dense rows show the unpruned baseline models. The $\Delta\%$ column shows the percentage change relative to the dense model. Bold values indicate best performance.

| Model | Method | Sparsity | FinMTEB | | | | | ChemTEB | | | |
|---|---|---|---|---|---|---|---|---|---|---|---|
| | | | Retr. | Cla. | STS | Avg. | $\Delta\%$ | Retr. | Cla. | Avg. | $\Delta\%$ |
| Qwen3-Embedding-4B | Dense | – | 0.6378 | 0.6100 | 0.3580 | 0.5353 | – | 0.6858 | 0.8419 | 0.7639 | – |
| | Random | 30% | 0.0111 | 0.3752 | 0.2354 | 0.2072 | -61.29% | 0.0000 | 0.4350 | 0.2175 | -71.52% |
| | Magnitude | 30% | 0.6263 | 0.6247 | 0.3736 | 0.5415 | +1.16% | **0.6851** | 0.8426 | 0.7639 | 0.00% |
| | General Fisher | 30% | 0.5850 | 0.6017 | **0.3807** | 0.5225 | -2.39% | 0.6327 | 0.8300 | 0.7313 | -4.26% |
| | Domain Fisher | 30% | 0.6245 | 0.6150 | 0.3311 | 0.5235 | -2.20% | 0.6843 | 0.8387 | 0.7615 | -0.30% |
| | **GAPrune (ours)** | 30% | **0.6278** | **0.6259** | 0.3739 | **0.5425** | **+1.35%** | 0.6845 | **0.8437** | **0.7641** | **+0.04%** |
| | Random | 50% | 0.0094 | 0.4028 | 0.2373 | 0.2165 | -59.55% | 0.0175 | 0.4714 | 0.2445 | -68.00% |
| | Magnitude | 50% | 0.5722 | 0.5984 | 0.3807 | 0.5171 | -3.40% | 0.6310 | 0.8289 | 0.7299 | -4.44% |
| | General Fisher | 50% | 0.1471 | 0.5732 | 0.3665 | 0.3623 | -32.32% | 0.4826 | 0.8096 | 0.6461 | -15.42% |
| | Domain Fisher | 50% | 0.5528 | 0.5911 | 0.3224 | 0.4887 | -8.70% | 0.5852 | 0.8268 | 0.7060 | -7.57% |
| | **GAPrune (ours)** | 50% | **0.5763** | **0.6067** | **0.3840** | **0.5224** | **-2.41%** | **0.6564** | **0.8360** | **0.7462** | **-2.31%** |
| E5-mistral-7B-Instruct | Dense | – | 0.6168 | 0.6450 | 0.3801 | 0.5473 | – | 0.3677 | 0.8276 | 0.5976 | – |
| | Random | 30% | 0.0433 | 0.5109 | 0.3125 | 0.2889 | -47.21% | 0.0322 | 0.6263 | 0.3292 | -44.91% |
| | Magnitude | 30% | 0.6148 | **0.6459** | **0.3791** | 0.5466 | -0.13% | 0.3750 | 0.8272 | 0.6011 | +0.58% |
| | General Fisher | 30% | 0.5998 | 0.5965 | 0.3775 | 0.5246 | -4.15% | 0.3640 | 0.8210 | 0.5925 | -0.86% |
| | Domain Fisher | 30% | 0.6145 | 0.6406 | 0.3694 | 0.5415 | -1.07% | 0.3789 | 0.8310 | 0.6050 | +1.22% |
| | **GAPrune (ours)** | 30% | **0.6162** | 0.6457 | 0.3790 | **0.5470** | **-0.06%** | **0.3800** | **0.8314** | **0.6057** | **+1.34%** |
| | Random | 50% | 0.0095 | 0.4212 | 0.2484 | 0.2264 | -58.64% | 0.0139 | 0.4923 | 0.2531 | -57.65% |
| | Magnitude | 50% | 0.6013 | 0.6316 | 0.3808 | 0.5379 | -1.72% | 0.4085 | 0.8294 | 0.6189 | +3.56% |
| | General Fisher | 50% | 0.5487 | 0.5965 | 0.3780 | 0.5077 | -7.23% | 0.3383 | 0.8192 | 0.5787 | -3.16% |
| | Domain Fisher | 50% | 0.5995 | 0.6290 | 0.3704 | 0.5330 | -2.61% | 0.3770 | 0.8287 | 0.6029 | +0.88% |
| | **GAPrune (ours)** | 50% | **0.6096** | **0.6387** | **0.3854** | **0.5446** | **-0.50%** | **0.4102** | **0.8309** | **0.6206** | **+3.84%** |

Our one-shot pruning experiments reveal several key insights about how different pruning strategies affect embedding model performance. At 30% sparsity, GAPrune outperforms all baselines on both benchmarks, achieving +1.35% improvement on FinMTEB and +0.04% on ChemTEB for Qwen3-Embedding-4B. This suggests that our DAI scoring successfully identifies parameters critical for domain-specific performance while removing those that create conflicts between general and domain objectives.

The performance differences become clearer at 50% sparsity. Random pruning causes severe degradation (40-60% drop), while GAPrune maintains performance within 2.5% of the dense model. More importantly, Fisher-based methods show larger drops than our approach, particularly General Fisher pruning, which degrades by over 30% on FinMTEB. This indicates that gradient alignment provides crucial information beyond what Fisher information alone can capture. We further explore the limits of our method in Appendix I, demonstrating that GAPrune maintains robust performance even at extreme sparsity levels (60% and 65%) where baseline methods suffer catastrophic collapse.

However, one-shot pruning only tells part of the story. In practice, pruned models are often retrained to recover performance, which raises the question of whether our pruning strategy provides a good foundation for post-pruning optimization.

## 4.3 PRUNE-AND-RETRAIN EVALUATION

The retraining experiments show that GAPrune not only recovers from pruning but often exceeds the original dense model performance. Qwen3-Embedding-4B improves by +4.51% on FinMTEB and +1.73% on ChemTEB after retraining, suggesting that our pruning removes redundant parameters while preserving the model's learning capacity for domain-specific tasks.

The comparison with $L^3$ Prune reveals an important distinction. While $L^3$ Prune also shows improvements after retraining, GAPrune consistently achieves higher performance across all metrics. This difference likely stems from our method's ability to preserve domain-specific knowledge during pruning, providing a better starting point for retraining. The gradient alignment component appears particularly crucial here, as it helps maintain the model's ability to capture specialized semantic patterns. These results hold across different architectures. E5-mistral-7B-Instruct shows

Table 2: Prune-and-Retrain Results on FinMTEB and ChemTEB Datasets (50% sparsity). The $\Delta\%$ column shows the percentage change relative to the dense model. Bold values indicate best performance.

| Model | Method | FinMTEB | | | | | ChemTEB | | | |
|---|---|---|---|---|---|---|---|---|---|---|
| | | Retr. | Cla. | STS | Avg. | $\Delta\%$ | Retr. | Cla. | Avg. | $\Delta\%$ |
| Qwen3-Embedding-4B | Dense | 0.6378 | 0.6100 | 0.3580 | 0.5353 | – | 0.6858 | 0.8419 | 0.7639 | – |
| | Magnitude | 0.6367 | 0.6382 | 0.3836 | 0.5528 | +3.28% | 0.6713 | 0.8317 | 0.7515 | -1.62% |
| | General Fisher | 0.6350 | 0.6317 | 0.3632 | 0.5433 | +1.49% | 0.6858 | 0.8345 | 0.7601 | -0.49% |
| | Domain Fisher | 0.6372 | 0.6353 | 0.3877 | 0.5534 | +3.39% | 0.6855 | 0.8347 | 0.7601 | -0.49% |
| | $L^3$ Prune | 0.6361 | 0.6328 | 0.3577 | 0.5422 | +1.29% | 0.6866 | 0.8418 | 0.7642 | +0.05% |
| | **GAPrune (ours)** | **0.6401** | **0.6402** | **0.3980** | **0.5594** | **+4.51%** | **0.7119** | **0.8422** | **0.7770** | **+1.73%** |
| E5-mistral-7B-Instruct | Dense | 0.6168 | 0.6450 | 0.3801 | 0.5473 | – | 0.3677 | 0.8276 | 0.5976 | – |
| | Magnitude | 0.5941 | 0.6704 | **0.3842** | 0.5496 | +0.41% | 0.4745 | 0.8267 | 0.6506 | +8.86% |
| | General Fisher | 0.5864 | 0.6713 | 0.3762 | 0.5446 | -0.49% | 0.4578 | 0.8319 | 0.6448 | +7.89% |
| | Domain Fisher | 0.5832 | 0.6718 | 0.3793 | 0.5448 | -0.46% | 0.4725 | 0.8295 | 0.6510 | +8.93% |
| | $L^3$ Prune | 0.6083 | 0.6710 | 0.3426 | 0.5406 | -1.23% | 0.5071 | 0.8302 | 0.6687 | +11.88% |
| | **GAPrune (ours)** | **0.6153** | **0.6721** | **0.3842** | **0.5572** | **+1.81%** | **0.5219** | **0.8327** | **0.6773** | **+13.33%** |

similar patterns, with GAPrune achieving the best performance on both datasets. The consistency suggests that our DAI-based approach captures fundamental principles of parameter importance that generalize across model architectures and domains. Additionally, we compare our InfoNCE retraining strategy with Knowledge Distillation in Appendix J, confirming that domain-specific retraining leverages the sparse architecture identified by GAPrune more effectively than standard distillation approaches. This difference arises may because KD forces the pruned model to mimic the general teacher, thereby treating the teacher's capabilities as an upper bound on domain adaptation. In contrast, direct retraining allows the model to learn specialized patterns that may be absent in the teacher's representations.

## 4.4 GEOMETRIC AND ADDITIONAL ANALYSIS

In this section, we further analyze the relationship between GAPrune and Fisher-based methods, demonstrating that our approach captures different parameter importance patterns and produces sparse embedding models with superior geometric properties.

### 4.4.1 LAYER CORRELATION ANALYSIS

To understand how different pruning methods prioritize parameters across model layers, we analyze the correlation between GAPrune and Fisher-based methods. As illustrated in the Appendix L, GAPrune shows minimal correlation with both Domain Fisher (-0.406) and General Fisher (-0.459) methods, indicating that our dual-criteria evaluation (domain importance and gradient alignment) identifies different parameters as critical compared to single-objective Fisher-based approaches. In contrast, Domain Fisher and General Fisher show high correlation (0.978) despite being computed on different datasets, suggesting that Fisher Information alone is relatively domain-agnostic and fails to distinguish between parameters important for general linguistic understanding versus those critical for domain-specific tasks.

Layer-wise analysis further demonstrates GAPrune's better understanding of parameter importance. By extracting hidden states from different layers of Qwen3-Embedding-4B and evaluating on FinMTEB tasks, we observe that retrieval performance improves significantly in the later layers (around layer 24), where high-level semantic representations are formed. However, Fisher-based methods prune more aggressively in these critical layers, removing parameters essential for retrieval performance. In contrast, GAPrune's gradient alignment component helps identify parameters that maintain both general semantic foundations and domain-specific patterns, leading to better retention of parameters in layers crucial for embedding quality. Detailed layer-wise importance scores and performance analysis are provided in Appendix M (Figure 2).

### 4.4.2 GEOMETRIC HYPERSPHERE ANALYSIS

We analyze several key metrics including uniformity loss (Wang & Isola, 2020), alignment loss (Wang & Isola, 2020), cross-dimensional correlation, effective dimensionality (Roy & Vetterli,

2007), and cosine similarity between pruned and dense embeddings to evaluate how pruning affects the geometric properties of embeddings. Our analysis shows that GAPrune achieves better geometric properties compared to baseline methods, demonstrating the best alignment between queries and positive samples (0.51 alignment loss) while maintaining high cross-dimensional correlation (0.52) and cosine similarity (0.22), indicating better preservation of semantic relationships. The detailed geometric analysis results and computing process are provided in Appendix N (Table 14).

## 5 CONCLUSION

In this work, we present GAPrune, a gradient-alignment pruning framework for domain-specific embedding models. Our key insight is that effective pruning requires understanding both parameter importance and cross-domain alignment, rather than treating all parameters uniformly. By combining Fisher Information with gradient alignment, GAPrune identifies parameters that are both critical for domain performance and well-aligned with general objectives. Experiments show that GAPrune maintains performance within 2.5% of dense models at 50% sparsity in one-shot pruning, while achieving performance improvements with retraining in 100 steps. These results hold across different model architectures and domains, demonstrating that principled domain-aware compression can achieve both efficiency and performance for specialized embedding models.

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

## A  ALGORITHM SUMMARY

Algorithm 1 presents the complete GAPrune procedure, which operates through three sequential stages: (1) representative sampling to distill essential statistical properties from both general and domain-specific corpora, (2) comprehensive parameter analysis that computes Fisher information and gradient alignment scores for each parameter, and (3) unified importance scoring and pruning based on our proposed Domain-Aware Importance (DAI) metric.

---

**Algorithm 1** Gradient-Alignment Pruning (GAPrune)

---

**Require:** Pre-trained embedding model $\mathcal{M}$ with parameters $\theta \in \mathbb{R}^d$, general corpus $\mathcal{D}_{gen}$, domain corpus $\mathcal{D}_{dom}$, target sparsity $s \in (0, 1)$, trade-off parameters $\alpha, \beta, \gamma \geq 0$
**Ensure:** Pruned model $\mathcal{M}_{pruned}$ with sparsity $s$
 1: **Stage 1: Representative Sampling**
 2: $\mathcal{S}_{gen} \leftarrow \text{KMeansSample}(\mathcal{D}_{gen}, k = 5000)$ {Sample representative general data}
 3: $\mathcal{S}_{dom} \leftarrow \text{KMeansSample}(\mathcal{D}_{dom}, k = 5000)$ {Sample representative domain data}
 4: **Stage 2: Parameter Analysis**
 5: **for** each parameter $\theta_j \in \theta$ **do**
 6:     Compute gradients: $g_j^{gen} \leftarrow \nabla_{\theta_j} \mathcal{L}(\mathcal{M}, \mathcal{S}_{gen})$
 7:     Compute gradients: $g_j^{dom} \leftarrow \nabla_{\theta_j} \mathcal{L}(\mathcal{M}, \mathcal{S}_{dom})$
 8:     Estimate Fisher info: $F_{jj}^{gen} \leftarrow \frac{1}{|\mathcal{S}_{gen}|} \sum_i (g_{j,i}^{gen})^2$
 9:     Estimate Fisher info: $F_{jj}^{dom} \leftarrow \frac{1}{|\mathcal{S}_{dom}|} \sum_i (g_{j,i}^{dom})^2$
10:     Compute alignment: $s_{g,j} \leftarrow \frac{\langle g_j^{gen}, g_j^{dom} \rangle}{\|g_j^{gen}\| \|g_j^{dom}\| + \varepsilon}$
11: **end for**
12: **Stage 3: DAI Scoring**
13: **for** each parameter $\theta_j \in \theta$ **do**
14:     Compute magnitude: $|\theta_j|$
15:     Compute DAI score: $\text{DAI}_j \leftarrow \left( (F_{jj}^{dom} - \beta \cdot F_{jj}^{gen}) \cdot |\theta_j| + \gamma \cdot \sqrt{|\theta_j|} \right) \cdot (1 + \alpha \cdot s_{g,j})$
16: **end for**
17: Sort parameters by DAI scores: $\{\theta_{(1)}, \theta_{(2)}, \ldots, \theta_{(d)}\}$
18: Retain top $(1 - s) \cdot d$ parameters: $\mathcal{M}_{pruned} \leftarrow \text{Mask}(\mathcal{M}, \{\theta_{(1)}, \ldots, \theta_{((1-s) \cdot d)}\})$
19: **return** $\mathcal{M}_{pruned}$

---

**Computational Complexity** The computation costs of GAPrune mainly come from the parameter analysis stage. Computing gradients and Fisher information requires one forward and backward pass over $N$ samples in each calibration set, resulting in $O(|\theta| \cdot N)$ complexity. The remaining operations, gradient alignment and DAI score computation, are simple element-wise vector operations with $O(|\theta|)$ cost. This single-shot analysis makes GAPrune practical for large-scale models without iterative procedures or retraining.

## B   EVALUATION INSTRUCTIONS

This section provides the evaluation instructions for all tasks used in our experiments. Table 3 shows the evaluation instructions for the 20 FinMTEB tasks used in our experiments. Table 4 shows the evaluation instructions for the 19 ChemTEB tasks used in our experiments.

## C   BENCHMARK METRICS

This section provides the detailed benchmark metrics for all tasks used in our experiments. As the domain benchmarks follow the MTEB evaluation protocol (Muennighoff et al., 2023), we use the following main metrics for each task type:

**Classification Tasks.** For classification tasks, we use accuracy as the primary metric, which measures the percentage of correctly classified samples.

**Retrieval Tasks.** For retrieval tasks, we use nDCG@10 (Normalized Discounted Cumulative Gain at rank 10) as the primary metric, which measures the quality of ranking by considering both relevance and position of retrieved documents.

**Semantic Textual Similarity (STS) Tasks.** For STS tasks, we use Spearman correlation as the primary metric, which measures the rank correlation between predicted similarity scores and ground truth similarity scores.

Table 3: Evaluation instructions for FinMTEB tasks

| Task | Instruction |
|------|-------------|
| **Retrieval Tasks (8)** | |
| FiQA2018Retrieval | Given a financial question, retrieve user replies that best answer the question. |
| FinanceBenchRetrieval | Given a financial question, retrieve the related context. |
| HC3Retrieval | Given a financial question, retrieve relevant passages that answer the query. |
| Apple10KRetrieval | Given a financial question, retrieve the related context. |
| TATQARetrieval | Given a financial question, retrieve user replies that best answer the question. |
| FinQARetrieval | Given a financial question, retrieve user replies that best answer the question. |
| USNewsRetrieval | Given a financial question, retrieve documents that can help answer the question. |
| TheGoldmanEnRetrieval | Given a financial term, retrieve the related context. |
| **Classification Tasks (8)** | |
| FinancialPhraseBankClassification | Classify the sentiment of a given finance text as either positive, negative, or neutral. |
| FinSentClassification | Classify the sentiment of a given finance text as either positive, negative, or neutral. |
| FiQAClassification | Perform aspect based financial sentiment classification. |
| SemEva2017Classification | Classify the sentiment of a given finance text as either positive, negative, or neutral. |
| FLSClassification | Classify the sentence into 'not-fls', 'specific fls', or 'non-specific fls' class. |
| ESGClassification | Classify the following sentence into one of the 'environmental','social', 'governance', 'non-esg' classes. |
| FOMCClassification | Classify the following sentence from FOMC into 'hawkish', 'dovish', or 'neutral' class. |
| FinancialFraudClassification | Detecting financial fraud from the given text. |
| **STS Tasks (2)** | |
| FinSTS | Detecting Subtle Semantic Shifts in Financial Narratives. |
| FINAL | Retrieve semantically similar finance text. |

## D  HYPERPARAMETERS IN RETRAINING EXPERIMENTS

For all prune-and-retrain experiments reported in Section 4.3 and Appendix J, we use consistent hyperparameters across different methods. We train all models using the AdamW optimizer with a learning rate of 1e-5 and linear learning rate scheduling. The weight decay is set to 0.01. We use a maximum sequence length of 512 and achieve an effective batch size of 512 through gradient accumulation over 16 steps with a per-device batch size of 4. All models are trained for 100 steps with bf16 precision on 8 NVIDIA H800 GPUs. These hyperparameters are kept constant across all retraining experiments to ensure fair comparison between different pruning methods and training strategies.

## E  RUNTIME ANALYSIS

To evaluate the computational efficiency of GAPrune, we compare its floating-point operations (FLOPs) against the dense Qwen3-Embedding-4B model on 4×3090 GPUs. As shown in Table 5, GAPrune achieves a 33.4% reduction in computational requirements. We also measure the actual runtime performance on the FiQARetrieval task, with each GPU consuming 10.0 GB of memory. The results are presented in Table 6.

Table 4: Evaluation instructions for ChemTEB tasks

| Task | Instruction |
|---|---|
| **Classification Tasks (17)** | |
| SDSEyeProtectionClassification | Classify whether eye protection is required when handling the chemical based on its safety data sheet. |
| SDSGlovesClassification | Classify whether gloves are required when handling the chemical based on its safety data sheet. |
| WikipediaBioMetChemClassification | Classify chemistry texts into biometallurgical chemistry or other chemistry fields. |
| WikipediaBiolumNeurochemClassification | Classify chemistry texts into bioluminescence/neurochemistry or other chemistry fields. |
| WikipediaChemEngSpecialtiesClassification | Classify chemistry texts into chemical engineering specialties or other chemistry fields. |
| WikipediaChemFieldsClassification | Classify chemistry texts into different main chemistry fields. |
| WikipediaChemistryTopicsClassification | Classify chemistry texts into specific chemistry topics or general chemistry. |
| WikipediaCompChemSpectroscopyClassification | Classify chemistry texts into computational chemistry/spectroscopy or other chemistry fields. |
| WikipediaCryobiologySeparationClassification | Classify chemistry texts into cryobiology/separation processes or other chemistry fields. |
| WikipediaCrystallographyAnalyticalClassification | Classify chemistry texts into crystallography/analytical chemistry or other chemistry fields. |
| WikipediaGreenhouseEnantiopureClassification | Classify chemistry texts into greenhouse chemistry/enantiopure compounds or other chemistry fields. |
| WikipediaIsotopesFissionClassification | Classify chemistry texts into isotopes/nuclear fission or other chemistry fields. |
| WikipediaLuminescenceClassification | Classify chemistry texts into luminescence chemistry or other chemistry fields. |
| WikipediaOrganicInorganicClassification | Classify chemistry texts into organic chemistry or inorganic chemistry. |
| WikipediaSaltsSemiconductorsClassification | Classify chemistry texts into salts/semiconductors chemistry or other chemistry fields. |
| WikipediaSolidStateColloidalClassification | Classify chemistry texts into solid-state/colloidal chemistry or other chemistry fields. |
| WikipediaTheoreticalAppliedClassification | Classify chemistry texts into theoretical chemistry or applied chemistry. |
| **Retrieval Tasks (2)** | |
| ChemHotpotQARetrieval | Given a chemical question, retrieve the related context. |
| ChemNQRetrieval | Given a chemical question, retrieve the related context. |

## F    ABLATION STUDY ON DAI SCORE COMPONENTS

To isolate the impact of each component in our Domain Alignment Importance (DAI) score, we evaluate the following configurations on Qwen3-Embedding-4B at 60% sparsity using the FinMTEB benchmark:

- **Dense:** The unpruned baseline model serving as the performance upper bound.
- **Fisher Diff:** Removes parameters based on the difference between domain and general Fisher information: $(F_{jj}^{\text{dom}} - \beta \cdot F_{jj}^{\text{gen}}) \cdot |\theta_j|$. This tests the effectiveness of naive general knowledge subtraction.
- **Fisher Diff w/ Compression:** Adds the magnitude-based regularization term to the Fisher Diff baseline: $(F_{jj}^{\text{dom}} - \beta \cdot F_{jj}^{\text{gen}}) \cdot |\theta_j| + \gamma \cdot \sqrt{|\theta_j|}$. This examines the impact of preserving parameters with large representational capacity.
- **Fisher Diff w/ Alignment:** Incorporates gradient alignment into the Fisher Diff baseline: $\left[ (F_{jj}^{\text{dom}} - \beta \cdot F_{jj}^{\text{gen}}) \cdot |\theta_j| \right] \cdot (1 + \alpha \cdot s_g^j)$. This evaluates the contribution of cross-domain alignment.
- **DAI (Full):** Our complete formulation combining all components, as defined in Eq. 5.

Table 7 summarizes the contribution of each component. Our observations are as follows:

**Fisher-based importance remains a strong signal but must be used cautiously.**

The Fisher Diff baseline underperforms relative to more advanced variants. This indicates that simply subtracting general-domain Fisher Information is insufficient for identifying domain-specific parameters and may inadvertently remove useful shared knowledge.

**Compression regularization exhibits clear task-dependent behavior.** Fisher Diff w/ Compression boosts STS performance (0.4147) but severely degrades retrieval (0.2850), suggesting that magnitude-oriented regularization alone cannot reliably preserve semantic distinctions required across diverse task types. This reinforces that compression should not be the primary driver of importance scoring.

**Gradient alignment is the dominant factor enabling stable gains.** Introducing gradient alignment (Fisher Diff w/ Alignment) yields consistent improvements across tasks, achieving the strongest performance among ablated variants. This demonstrates that alignment effectively filters out parameters beneficial to both general and domain objectives, mitigating the pitfalls of naive Fisher subtraction.

**The full DAI formulation combines complementary strengths.** GAPrune (Full) achieves the best scores across all task categories, recovering 97.0% of dense performance despite 60% sparsity. The strong gains over Fisher w/ Alignment (e.g., +6.3% in retrieval and +4.5% in classification) confirm that Fisher subtraction, compression regularization, and gradient alignment interact synergistically rather than redundantly. Together, they form a robust criterion that consistently preserves domain-relevant capacity under heavy sparsification.

## G    ABLATION STUDY ON SAMPLE DATA SIZE

We investigate the sensitivity of our DAI scoring mechanism to the size of the calibration dataset. While our main experiments use 5,000 samples per domain, reducing this requirement could further improve computational efficiency. We vary the calibration data size from 25% (1,250 samples) to 100% (5,000 samples) and evaluate one-shot pruning performance on Qwen3-Embedding-4B at 50% sparsity.

Table 5: Computational Comparison.

| Model | FLOPs | Reduction |
|---|---|---|
| Dense | 8.24T | – |
| GAPrune | 5.48T | 33.4% |

Table 6: FiQARetrieval Test Runtime

| Model | Time (hours) |
|---|---|
| Dense | 1.89 |
| GAPrune | 1.17 |

Table 7: Ablation study results on FinMTEB across different task types at 60% sparsity. Best results in **bold**, second best underlined. The Avg. column shows the macro-average across all three task types.

| Method | Retr. | Cla. | STS | Avg. |
|---|---|---|---|---|
| Dense | 0.6378 | 0.6100 | 0.3580 | 0.5353 |
| Fisher Diff | 0.4630 | 0.4717 | 0.3161 | 0.4169 |
| Fisher Diff w/ Compression | 0.2850 | 0.5574 | **0.4147** | 0.4190 |
| Fisher Diff w/ Alignment | 0.5084 | 0.5758 | 0.3698 | 0.4847 |
| GAPrune (Full) | **0.5406** | **0.6020** | 0.4146 | **0.5191** |

Table 8: One-shot pruning performance on FinMTEB at 50% sparsity with varying calibration data sizes. Baseline uses 5,000 samples per domain (100%).

| Sample Data | Retr. | Cla. | STS | Avg. |
|---|---|---|---|---|
| 100% (5,000 samples) | 0.5763 | 0.6067 | 0.3840 | 0.5224 |
| 75% (3,750 samples) | 0.5887 | 0.6116 | 0.3858 | 0.5287 |
| 50% (2,500 samples) | 0.5879 | 0.6127 | 0.3875 | 0.5294 |
| 25% (1,250 samples) | 0.5891 | 0.6124 | 0.3872 | 0.5296 |

As shown in Table 8, GAPrune maintains robust performance across all tested sizes. Notably, using fewer samples (25%–75%) yields performance comparable to, or marginally better than, the full 5,000-sample baseline. This stability suggests that the statistical signals captured by Fisher Information and gradient alignment are consistent even with smaller data subsets, potentially reducing the risk of overfitting to specific calibration samples. These findings indicate that practitioners can effectively apply GAPrune with as few as 1,250 samples per domain, significantly reducing the computational overhead of the analysis phase.

## H  HYPERPARAMETER SENSITIVITY ANALYSIS

We analyze the robustness of GAPrune to variations in its three key hyperparameters: $\alpha$ (gradient alignment weight), $\beta$ (general knowledge subtraction weight), and $\gamma$ (compression regularization weight). We test each hyperparameter individually on FinMTEB at 50% sparsity while holding the others at their baseline values ($\alpha = 0.2$, $\beta = 1.0$, $\gamma = 0.5$).

Table 9: Hyperparameter sensitivity analysis on FinMTEB at 50% sparsity. Baseline configuration: $\alpha = 0.2$, $\beta = 1.0$, $\gamma = 0.5$.

| Configuration | Retr. | Cla. | STS | Avg. |
|---|---|---|---|---|
| *Gradient Alignment Weight ($\alpha$)* | | | | |
| $\alpha = 0.1$ | 0.5895 | 0.6148 | 0.3879 | 0.5257 |
| $\alpha = 0.2$ | 0.5763 | 0.6067 | 0.3840 | 0.5224 |
| $\alpha = 0.3$ | 0.5894 | 0.6147 | 0.3877 | 0.5256 |
| *General Knowledge Subtraction Weight ($\beta$)* | | | | |
| $\beta = 0.75$ | 0.5900 | 0.6149 | 0.3886 | 0.5262 |
| $\beta = 1.0$ | 0.5763 | 0.6067 | 0.3840 | 0.5224 |
| $\beta = 1.25$ | 0.5897 | 0.6150 | 0.3891 | 0.5308 |
| *Compression Regularization Weight ($\gamma$)* | | | | |
| $\gamma = 0.3$ | 0.5898 | 0.6150 | 0.3879 | 0.5259 |
| $\gamma = 0.5$ | 0.5763 | 0.6067 | 0.3840 | 0.5224 |
| $\gamma = 0.7$ | 0.5899 | 0.6151 | 0.3890 | 0.5308 |

Table 9 demonstrates that GAPrune is highly stable across a wide range of configurations. Gradient Alignment ($\alpha$): Performance remains consistent across $\alpha \in \{0.1, 0.2, 0.3\}$, with minor fluctuations

($< 0.5\%$), confirming that the method is not overly sensitive to the exact strength of the alignment signal. Subtraction Weight ($\beta$): Increasing $\beta$ to 1.25 yields a slight improvement in STS tasks, but the baseline $\beta = 1.0$ offers the most balanced performance across retrieval and classification. Regularization Weight ($\gamma$): Similar to $\beta$, variations in $\gamma$ result in negligible performance shifts. Overall, these results confirm that the baseline configuration is robust and does not require extensive tuning for different deployments.

## I  ONE-SHOT PRUNING ON HIGHER SPARSITY

To test the limits of our pruning strategy, we evaluate GAPrune at aggressive sparsity levels (60% and 65%) on both FinMTEB and ChemTEB benchmarks.

Table 10: One-shot pruning results on FinMTEB and ChemTEB at 60% and 65% sparsity. GAPrune maintains substantial performance advantages over baselines even at extreme sparsity levels where other methods collapse.

| Method | Sparsity | FinMTEB | | | | ChemTEB | | |
|---|---|---|---|---|---|---|---|---|
| | | Retr. | Cla. | STS | Avg. | Retr. | Cla. | Avg. |
| Dense | – | 0.6378 | 0.6100 | 0.3580 | 0.5353 | 0.6858 | 0.8419 | 0.7639 |
| Magnitude | 60% | 0.2818 | 0.5552 | 0.3884 | 0.4085 | 0.3101 | 0.8071 | 0.5586 |
| General Fisher | 60% | 0.2925 | 0.4808 | 0.2852 | 0.3528 | 0.0816 | 0.6066 | 0.3441 |
| Domain Fisher | 60% | 0.4914 | 0.5636 | 0.3227 | 0.4592 | 0.3866 | 0.8155 | 0.6011 |
| GAPrune | 60% | **0.5406** | **0.6020** | **0.4146** | **0.5191** | **0.5980** | **0.8334** | **0.7157** |
| Magnitude | 65% | 0.0250 | 0.4339 | 0.2652 | 0.2414 | 0.0038 | 0.6436 | 0.3237 |
| General Fisher | 65% | 0.0243 | 0.4133 | 0.2237 | 0.2204 | 0.0424 | 0.4601 | 0.2513 |
| Domain Fisher | 65% | 0.4310 | 0.5795 | 0.3199 | 0.4435 | 0.0278 | 0.4588 | 0.2433 |
| GAPrune | 65% | **0.4607** | **0.5906** | **0.4274** | **0.4929** | **0.4231** | **0.8197** | **0.6214** |

Table 10 demonstrates that GAPrune maintains strong performance at aggressive sparsity levels where baseline methods fail.

**Robustness against collapse.** At 65% sparsity, baseline methods (Magnitude and General Fisher) suffer catastrophic degradation, with retrieval performance dropping to near-zero levels (e.g., 0.0250 on FinMTEB). In contrast, GAPrune maintains a respectable average score of 0.4929 on FinMTEB.

**Domain specialization at the limit.** The advantage is particularly evident on ChemTEB. At 65% sparsity, GAPrune retains 81.3% of the dense baseline's performance (0.6214 vs 0.7639), whereas the strongest baseline, Domain Fisher, collapses to 0.2433.

**Preservation of Retrieval Capabilities.** Retrieval tasks are most sensitive to pruning. GAPrune's ability to maintain high retrieval scores (0.4607 on FinMTEB at 65% sparsity) vs. the baselines' failure indicates that our DAI score successfully identifies the "long-tail" parameters essential for semantic matching, which other methods discard.

## J  PRUNE-AND-RETRAIN WITH KNOWLEDGE DISTILLATION

We investigate the interplay between pruning quality and post-pruning optimization strategies. Specifically, we examine whether Knowledge Distillation (KD) from the dense teacher model can recover performance for randomly pruned models versus those pruned with GAPrune. We compare three strategies on Qwen3-Embedding-4B (50% sparsity, FinMTEB): (1) One-shot pruning (no retraining), (2) Retraining with KD ($\mathcal{L}_{KD} = \text{MSE}(f_{teacher}, f_{student})$), and (3) Retraining with InfoNCE on domain data.

**Experimental Setup.** We compare three post-pruning strategies: no retraining (one-shot pruning), retraining with KD, and retraining with InfoNCE loss on domain data. All experiments use Qwen3-Embedding-4B at 50% sparsity on FinMTEB with 100 training steps, batch size of 512, and learning rate of 1e-5. For knowledge distillation, we follow the sentence embedding distillation approach

(Reimers & Gurevych, 2020), using the unpruned Qwen3-Embedding-4B as the teacher model. The distillation loss minimizes the mean squared error between the teacher and student embeddings:

$$\mathcal{L}_{\mathrm{KD}} = \mathrm{MSE}(f_{\mathrm{teacher}}(x), f_{\mathrm{student}}(x))$$

Table 11: Prune-and-retrain results with different training strategies on FinMTEB at 50% sparsity. InfoNCE retraining on domain data outperforms knowledge distillation from the dense model.

| Method | Retr. | Cla. | STS | Avg. |
|---|---|---|---|---|
| Dense | 0.6378 | 0.6100 | 0.3580 | 0.5353 |
| Random Prune (one-shot) | 0.0094 | 0.4028 | 0.2373 | 0.2165 |
| Random Prune + KD | 0.0095 | 0.4032 | 0.2070 | 0.2066 |
| GAPrune (one-shot) | 0.5763 | 0.6067 | 0.3840 | 0.5224 |
| GAPrune + KD | 0.5883 | 0.6088 | 0.3816 | 0.5262 |
| GAPrune + InfoNCE | **0.6401** | **0.6402** | **0.3980** | **0.5594** |

The results in Table 11 reveal several critical insights:

**Pruning quality limits KD effectiveness.** KD fails to rescue a randomly pruned model (0.2066 average), performing slightly worse than the one-shot random baseline. This confirms that if the structural foundation is destroyed by poor pruning, distillation cannot compensate for the loss.

**GAPrune provides a distinct architectural advantage.** GAPrune combined with KD improves performance to 0.5262, recovering 98.3% of the dense model's capability. This suggests that GAPrune preserves the specific parameters necessary for effective knowledge transfer.

**Domain-specific retraining outperforms distillation.** Retraining GAPrune with InfoNCE on domain data yields the highest performance (0.5594), surpassing the original dense model (+4.5%). This indicates that the sparse architecture identified by GAPrune is not merely a compressed version of the teacher, but a highly efficient structure primed for domain specialization.

## K  DOMAIN-DEPENDENT PARAMETER ANALYSIS

A core premise of our method is that parameter importance is not static but varies by domain. We empirically validate this by analyzing the Fisher Information scores of Qwen3-Embedding-4B parameters across Finance, Chemistry, and General domains. We compute the Spearman rank correlation of parameter importance between domain pairs and calculate the percentage of parameters that experience a "significant rank shift" (defined as a change in importance percentile $> 20\%$).

Table 12 confirms domain divergence. The moderate correlations (0.62–0.67) imply that a parameter critical for the General domain is not necessarily critical for Finance or Chemistry. Notably, over one-third (32.4%–35.9%) of parameters undergo large rank shifts when switching domains. This supports the necessity of our gradient alignment approach, which specifically targets parameters that maintain consistent behavior across domains while safely removing those that create conflicts.

Table 12: Spearman rank correlation of parameter importance across domains and percentage of parameters with large rank shifts ($> 20$ percentile change). All correlations are significant at $p < 0.001$ based on 1M sampled parameters.

| Domain Pair | Spearman $\rho$ | Large Rank Shifts |
|---|---|---|
| Finance - General | 0.627 | 32.4% |
| Chemistry - General | 0.675 | 33.6% |
| Chemistry - Finance | 0.674 | 35.9% |

## L    METHOD CORRELATION ANALYSIS

This section analyzes the correlation between different pruning methods to understand how they rank parameters differently.

For each method, we first normalize the importance scores to rank scores (ranging from 0 to 1) based on the parameter's relative importance within that method. We then compute Pearson correlation coefficients between the normalized rank scores of different methods across all common parameters.

Table 13: Correlation matrix between different pruning methods based on normalized rank scores. Values range from -1 (perfect negative correlation) to +1 (perfect positive correlation).

| Method | Correlation with | | |
|---|---|---|---|
| | **GAPrune** | **Domain Fisher** | **General Fisher** |
| GAPrune | 1.000 | -0.406 | -0.459 |
| Domain Fisher | -0.406 | 1.000 | 0.978 |
| General Fisher | -0.459 | 0.978 | 1.000 |

The correlation analysis reveals important insights about the relationship between different pruning approaches. GAPrune shows negative correlations with both Fisher-based methods (-0.406 with Domain Fisher and -0.459 with General Fisher). This suggests that GAPrune provides complementary information beyond what Fisher Information alone can capture.

The high positive correlation between Domain Fisher and General Fisher (0.978) indicates that these methods rank parameters very similarly, despite being computed on different datasets. This also suggests that Fisher Information may be relatively domain-agnostic, which could explain why both methods struggle to preserve domain-specific knowledge during pruning.

## M    LAYER ANALYSIS

This section provides detailed analysis of layer-wise parameter importance and performance across different pruning methods.

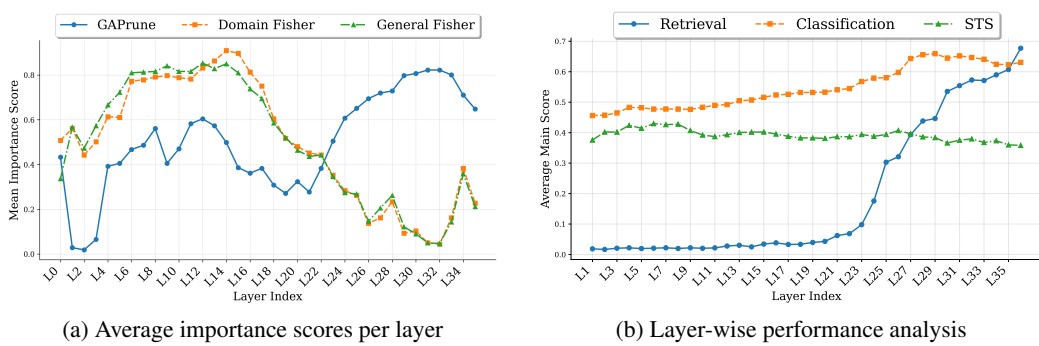

(a) Average importance scores per layer                 (b) Layer-wise performance analysis

Figure 2: Layer-wise analysis of pruning methods. (a) Average importance scores per layer indicate that GAPrune assigns higher importance to parameters in critical later layers compared to Fisher-based methods. (b) Performance across different layers shows that retrieval tasks benefit significantly from later layers.

**Layer-wise Importance Analysis.** Figure 2a shows the average importance scores assigned by different pruning methods across all layers of Qwen3-Embedding-4B. The analysis reveals several key patterns: (1) GAPrune consistently assigns higher importance scores to parameters in the later layers (layers 20-32), particularly around layer 24-28, where high-level semantic representations are typically formed. (2) Fisher-based methods (both Domain Fisher and General Fisher) show more uniform importance distribution across layers, with slightly higher scores in the middle layers but

lower scores in the critical later layers. (3) This difference in layer-wise importance assignment explains why GAPrune better preserves retrieval performance, as it retains more parameters in the layers most crucial for semantic understanding.

**Layer-wise Performance Analysis.** Figure 2b demonstrates the performance of different layers when used as final representations for FinMTEB retrieval tasks. The analysis shows that: (1) Retrieval performance improves significantly in the later layers, with the most substantial gains occurring around layer 28. (2) Early layers (1-21) show relatively poor retrieval performance, as they primarily capture low-level linguistic features. (3) Middle layers (23-27) show gradual improvement, indicating the development of more complex semantic representations. (4) The performance pattern aligns with GAPrune's importance assignment strategy, confirming that our method correctly identifies and preserves parameters in the most critical layers for embedding quality.

## N    GEOMETRIC METRICS COMPUTATION

To understand how pruning affects the geometric properties of embeddings, we analyze several key metrics that capture different aspects of embedding quality. We evaluate uniformity loss (Wang & Isola, 2020) to measure how uniformly distributed embeddings are in the embedding space, alignment loss (Wang & Isola, 2020) to assess how well query embeddings align with their positive samples, cross-dimensional correlation to quantify the dimensional relationships between query and positive embeddings, effective dimensionality (Roy & Vetterli, 2007) to measure how many dimensions are actually utilized in the embedding space, and cosine similarity to measure how well the pruned model preserves the original embedding structure. These metrics are computed on a sample of 1000 triplets from the finance domain dataset using Qwen3-Embedding-4B with 50% sparsity.

**Uniformity Loss.** The uniformity loss (Wang & Isola, 2020) measures how uniformly distributed the embeddings are in the embedding space. For a set of embeddings $\{z_i\}_{i=1}^n$, we compute:

$$\mathcal{L}_{\text{uniformity}} = \log \mathbb{E}_{i,j} \left[ \exp(-t\|z_i - z_j\|_2^2) \right] \tag{6}$$

where $t$ is a temperature parameter (set to 2.0) and the expectation is taken over all pairs $(i, j)$ with $i \neq j$. Lower values indicate a more uniform distribution.

**Alignment Loss.** The alignment loss (Wang & Isola, 2020) measures how well aligned query embeddings are with their corresponding positive samples. For query embeddings $\{q_i\}_{i=1}^n$ and positive embeddings $\{p_i\}_{i=1}^n$, we compute:

$$\mathcal{L}_{\text{alignment}} = \mathbb{E}_i \left[ \|q_i - p_i\|_2^\alpha \right] \tag{7}$$

where $\alpha$ is a power parameter (set to 2.0). Lower values indicate better alignment between queries and positives.

**Cross-Dimensional Correlation.** We compute the cross-dimensional correlation between query and positive embeddings by measuring the average absolute correlation across all dimensions:

$$\text{CDC} = \frac{1}{d} \sum_{k=1}^d |\text{corr}(q_{\cdot k}, p_{\cdot k})| \tag{8}$$

where $d$ is the embedding dimension, $q_{\cdot k}$ and $p_{\cdot k}$ are the $k$-th dimensions of all query and positive embeddings respectively, and $\text{corr}(\cdot, \cdot)$ is the Pearson correlation coefficient. This metric quantifies how well the pruned model preserves the dimensional relationships between queries and their positive samples.

**Effective Dimensions.** The effective dimensionality (Roy & Vetterli, 2007) measures how many dimensions are actually needed to capture the essential information in the embedding space. For embeddings $\{z_i\}_{i=1}^n$ with dimension $d$, we compute:

$$\text{Eff. Dim.} = \min_{k} \left\{ k : \frac{\sum_{i=1}^{k} \sigma_{(i)}^2}{\sum_{i=1}^{d} \sigma_{(i)}^2} \geq 0.95 \right\} \quad (9)$$

where $\sigma_{(1)}^2 \geq \sigma_{(2)}^2 \geq \ldots \geq \sigma_{(d)}^2$ are the sorted variances of each dimension, with $\sigma_k^2 = \text{Var}(z_{.k})$. Higher values indicate better utilization of the embedding space.

**Cosine Similarity.** We compute the average cosine similarity between embeddings from the pruned model and the dense model to measure how well the pruned model preserves the original embedding structure. For pruned embeddings $\{z_i^{pruned}\}_{i=1}^{n}$ and dense embeddings $\{z_i^{dense}\}_{i=1}^{n}$, we compute:

$$\text{Cosine Sim.} = \frac{1}{n} \sum_{i=1}^{n} \frac{z_i^{pruned} \cdot z_i^{dense}}{\|z_i^{pruned}\|_2 \|z_i^{dense}\|_2} \quad (10)$$

Higher values indicate that the pruned model better preserves the original embedding structure.

**Embedding Analysis Results.** As shown in Table 14, GAPrune achieves the best alignment between queries and positive samples (0.51 alignment loss) while maintaining high cross-dimensional correlation (0.52) and cosine similarity (0.22), indicating better preservation of semantic relationships. Magnitude pruning matches the dense model's uniformity (-3.30) but suffers from poor semantic alignment, with cross-dimensional correlation dropping to 0.44 and cosine similarity to 0.16. GAPrune uses 1820 out of 2560 dimensions compared to 1605 for domain Fisher pruning, suggesting that domain-only approaches prune too aggressively and remove parameters important for general knowledge. These results confirm that GAPrune's balanced approach maintains both statistical distribution and semantic structure.

Table 14: Embedding Analysis Results on Qwen3-Embedding-4B (Sample Size: 1000). Lower values are better for Uniformity Loss and Alignment Loss ($\downarrow$). Higher values are better for Mutual Info., Cosine Sim., and Effective Dim. ($\uparrow$). Bold values indicate best performance.

| Model | Uniformity Loss $\downarrow$ | Alignment Loss $\downarrow$ | Cross-Dim. Corr. $\uparrow$ | Cosine Sim. $\uparrow$ | Effective Dim. (out of 2560) |
|---|---|---|---|---|---|
| Dense | -3.30 | 0.79 | 0.54 | – | 2560 |
| Magnitude | **-3.30** | 0.79 | 0.44 | 0.16 | 1713 |
| General Fisher | -1.93 | 0.59 | 0.45 | 0.11 | 1715 |
| Domain Fisher | -2.82 | 0.75 | 0.51 | 0.14 | 1605 |
| GAPrune (ours) | -2.41 | **0.51** | **0.52** | **0.22** | **1820** |

