# OpenReview forum: "GAPrune: Gradient-Alignment Pruning for Domain-Aware Embeddings"
_ICLR.cc/2026/Conference — ICLR 2026 Poster_

### Official Review · Reviewer_qnve · 2025-11-01

**Soundness:** 3
**Presentation:** 4
**Contribution:** 3
**Rating:** 6
**Confidence:** 3

**Summary:**

This paper presents a method for conducting domain-aware pruning for embedding models.The method comes in three different stages: (1) gathering representative data points from general domain dataset and domain-specific dataset using k-means sampling ($k = 5000$); (2) calculating parameter importance as measured by gradient over InfoNCE loss on general domain samples and domain-specific samples, as well as gradient alignment between the general and specific domain; (3) computing a domain-aware importance (DAI) score using the importance and alignment measures above, then prune parameters based on the DAI score. Experimental results show better performance compared to baselines such as vanilla dense embedding and magnitude pruning, and better compatibility with domain-specific re-training after pruning.

**Strengths:**

1. The paper is very well-written. The approach has many components, but all the steps are very well explained and mostly well-motivated.
2. The method achieved significantly better performance over strong baselines under the re-training setup.

**Weaknesses:**

1. Under the setup without re-training, the performance improvement over more simplistic methods like magnitude pruning is minimal, so I'm not entirely convinced if the proposal is really worth it when no re-training is performed.
2. The necessity of each steps would be better justified if more rigorous ablation is performed. For example, while I understand the motivation of the gradient alignment term, I'm not sure if pruning "domain specific" parameters serves the stated goal of domain-aware embedding pruning. In defense of this proposal, this pruning choice might carry some benefit when re-training is performed (because re-training can recover domain-specific knowledge), but this needs to be validated by empirical experiments.

**Questions:**

Can the authors further comment on the motivation behind the $(1 + \alpha s^j_g)$ term? I might have missed or misunderstood some details.

Minor nit comment: L629 in Algorithm 1 seems like a self-assignment? I think you can either get rid of this line, or assign the magnitude to a variable.

---

> ### Author Response · Authors · 2025-11-17
> **Response to Reviewer qnve [1/3]**
>
> We sincerely thank Reviewer qnve for the positive assessment and the constructive feedback. We address each concern below with new experiments.
>
> `W1: Minimal Improvement Without Retraining`
>
> We appreciate this observation. At moderate sparsity (50%) in Table 1, the performance difference may appear modest. To address this concern, we evaluate at higher sparsity levels (60% and 65%) where the quality of pruning decisions becomes more critical and performance differences are amplified. These aggressive sparsity settings serve as a stress test to demonstrate the robustness of our approach:
>
>
> **FinMTEB  (Based on Qwen3-Embedding-4B) :**
>
> | Method | Sparsity | Retr. | Cla. | STS | Avg. |
> |--------|----------|-------|------|-----|------|
> | Dense | - | 0.6378 | 0.6100 | 0.3580 | 0.5353 |
> | Magnitude | 60% | 0.2818 | 0.5552 | 0.3884 | 0.4085 (-24%) |
> | GAPrune | 60% | **0.5406** | **0.6020** | **0.4146** | **0.5191** (-3%) |
> | Magnitude | 65% | 0.0250 | 0.4339 | 0.2652 | 0.2414 (-55%) |
> | GAPrune | 65% | **0.4607** | **0.5906** | **0.4274** | **0.4929** (-8%) |
>
> **ChemTEB  (Based on Qwen3-Embedding-4B) :**
>
> | Method | Sparsity | Retr. | Cla. | Avg. |
> |--------|----------|-------|------|------|
> | Dense | - | 0.6858 | 0.8419 | 0.7639 |
> | Magnitude | 60% | 0.3101 | 0.8071 | 0.5586 (-27%) |
> | GAPrune | 60% | **0.5980** | **0.8334** | **0.7157** (-6%) |
> | Magnitude | 65% | 0.0038 | 0.6436 | 0.3237 (-58%) |
> | GAPrune | 65% | **0.4231** | **0.8197** | **0.6214** (-19%) |
>
> At 65% sparsity, magnitude pruning collapses to 45% of the dense baseline on FinMTEB and 42% on ChemTEB, while GAPrune maintains 92% and 81% respectively. On ChemTEB, magnitude's retrieval drops to 0.0038 (near-random) while GAPrune achieves 0.4231. This demonstrates our method's value for aggressive compression scenarios where pruning decisions matter most.

---

> ### Author Response · Authors · 2025-11-17
> **Response to Reviewer qnve [2/3]**
>
> `W2: Necessity of Each Step`
>
> Thank you for raising this important point. We appreciate the opportunity to clarify our methodology and provide additional empirical validation.
>
> **Clarification on Pruning Strategy:**
>
> We would like to respectfully clarify a key aspect of our method: GAPrune does not primarily prune "domain-specific" parameters. Instead, our gradient alignment mechanism is designed to identify and preserve parameters that contribute positively to both the general and domain objectives. The parameters we remove are those that are either domain-agnostic (less critical for either) or exhibit strong conflicts (negatively impacting the domain task). This ensures the core, transferable knowledge is retained.
>
> **Component-Level Ablation:**
>
> To rigorously validate the contribution of each component, we conducted a comprehensive ablation study on Qwen3-Embedding-4B at 60% sparsity. The results are summarized below:
>
> | Method | Components | Retr. | Cla. | STS | Avg. |
> |--------|------------|-------|------|-----|------|
> | Fisher Diff | $(F^d - β·F^g)·\|θ_j\|$ | 0.4630 | 0.4717 | 0.3161 | 0.4169 (-9.2%) |
> | Fisher Diff w/ Compression | $+ γ·√\|θ_j\|$ | 0.2850 | 0.5574 | 0.4147 | 0.4190 |
> | Fisher Diff w/ Alignment | $+ (1 + α·s_g)$ | 0.5084 | 0.5758 | 0.3698 | 0.4847 (+18.6%) |
> | GAPrune (Full) | Complete DAI | 0.5406 | 0.6020 | 0.4146 | **0.5191** |
>
> **Key findings:** (1) Naively subtracting general Fisher (Row 1) underperforms, demonstrating that blindly removing "general" parameters is harmful. (2) Adding gradient alignment (Row 3: 0.4847) provides substantial improvement, identifying parameters that serve both general and domain objectives. (3) The complete DAI formulation (Row 4: 0.5191) achieves 97% of dense baseline performance while removing 60% of parameters, validating the necessity of all components working together.
>
> **Retraining Benefits and Architecture Quality:**
>
> The reviewer correctly notes that retraining can recover domain-specific knowledge. To address the core of your question—whether our one-shot pruning strategy correctly identifies which parameters to keep for the domain task, we designed a controlled experiment. We posit that if our one-shot pruning successfully preserves the crucial parameters for the domain, then the model before retraining should already possess strong domain capability, and retraining would only require minor recovery. Conversely, a poor pruning strategy would destroy the architecture's foundational capability, making recovery difficult.
>
> We compare three architectures (Dense, Random Prune, GAPrune) with two retraining methods (recovery-oriented KD vs. task-oriented InfoNCE) on Qwen3-Embedding-4B at 50% sparsity:
>
> | Architecture | Retraining | Retr. | Cla. | STS | Avg. | Δ from one-shot |
> |--------------|------------|-------|------|-----|------|-----------------|
> | Dense | - | 0.6378 | 0.6100 | 0.3580 | 0.5353 | - |
> | Random Prune | one-shot | 0.0094 | 0.4028 | 0.2373 | 0.2165 | - |
> | Random Prune | KD | 0.0095 | 0.4032 | 0.2070 | 0.2066 | +0.01 |
> | Random Prune | InfoNCE | 0.0127 | 0.4156 | 0.2401 | 0.2228 | +0.63 |
> | GAPrune | one-shot | 0.5763 | 0.6067 | 0.3840 | 0.5224 | - |
> | GAPrune | KD | 0.5883 | 0.6088 | 0.3816 | 0.5262 | +0.38 |
> | **GAPrune** | **InfoNCE** | **0.6401** | **0.6402** | **0.3980** | **0.5594** | **+3.70** |
>
>
> **Key Findings:**
>
> 1. **High One-shot Performance Confirms Correct Parameter Selection**: GAPrune achieves 97.6% of dense model performance before retraining. This demonstrates our method successfully preserves parameters critical for the domain, directly validating our pruning strategy.
>
> 2. **Quality Pruning Enables Effective Recovery**: The potential for recovery depends entirely on initial pruning quality. GAPrune shows significant gains with retraining, while random pruning fails to recover, proving our architecture is fundamentally sound.
>
> 3. **Optimized Architecture Surpasses Dense Baseline**: GAPrune with InfoNCE retraining exceeds the original dense model's performance. This shows our pruning creates a more efficient architecture that is better suited for domain adaptation.
>
> In summary, the high one-shot performance directly confirms the accuracy of our parameter selection, and the retraining results further demonstrate the architectural advantages it creates.
>
> ---
>
> Reimers, Nils, and Iryna Gurevych. "Making Monolingual Sentence Embeddings Multilingual using Knowledge Distillation." Proceedings of the 2020 Conference on Empirical Methods in Natural Language Processing (EMNLP). 2020.

---

> ### Author Response · Authors · 2025-11-17
> **Response to Reviewer qnve [3/3]**
>
> `Q1: Motivation for $(1 + α·s_g)$ Term`
>
> Thank you for this question. The gradient alignment term $(1 + α·s_g)$ implements our core insight: preserving parameters that contribute positively to both general and domain objectives leads to better pruning outcomes (Zhang et al., 2024).
>
> The term works as follows:
> - When $s_g > 0$: Parameter's gradients align between general and domain tasks → boost its importance score → preserve
> - When $s_g < 0$: Parameter's gradients conflict between tasks → reduce its importance score → may prune
> - When $s_g ≈ 0$: Parameter is task-agnostic → importance determined by Fisher Information alone
>
> This directly addresses your W2 concern: we don't blindly preserve all domain-specific parameters. Instead, we selectively preserve those that align with general capabilities, ensuring the pruned model maintains a strong foundation for domain adaptation.
>
> Our ablation validates this design: Fisher w/ Alignment achieves 0.4847 better than the Domain Fisher (0.4592) only method, demonstrating that prioritizing aligned parameters is more effective than preserving domain-important parameters indiscriminately.
>
>
> `Minor Comment: Algorithm 1 Line 629`
>
> Thank you for catching this. We fixed the self-assignment in the revision.
>
> ---
>
> Thank you for the thoughtful feedback. The new high-sparsity experiments and retraining analyses have substantially strengthened our work. We hope these additions address your concerns and welcome any further questions.
>
> ---
>
> Zhang, Nan, et al. "Pruning as a Domain-specific LLM Extractor." *Findings of the Association for Computational Linguistics: NAACL 2024*. 2024.

---

> ### Comment · Reviewer_qnve · 2025-11-27
>
> I thank the authors for the very thorough response. It addressed most of my concerns. It's remains somewhat unsatisfying to me that when no re-training is involved, the method only shows significant benefit over the very simple magnitude pruning at 65% sparsity (concurred by Reviewer nY2B).
>
> I read the other reviewers' comments and discussions, Reviewer 8Ku6 raised some good points such as reproducibility and lack of comparison with KD, but I don't think they fully overshadow the strengths of the paper listed in my review. I'm maintaining my score for now.

---

> ### Author Response · Authors · 2025-11-28
> **Response to Reviewer qnve**
>
> We appreciate that our response addressed most concerns. Regarding the remaining point about moderate improvement at 50% sparsity without retraining, we agree that this is expected: at moderate sparsity, most methods retain sufficient parameters, and pruning decisions matter less. Our method is most valuable when compression is aggressive or when preserving specific capabilities is critical.
>
> To further demonstrate GAPrune's advantage, we conducted an additional experiment on a different scenario. Our main experiments start from general pre-trained models because this reflects the typical workflow: practitioners take state-of-the-art general models (e.g., Qwen3-Embedding, E5-Mistral) and adapt them to specific domains.
>
> Here, we consider the complementary case: practitioners may already have a domain-adapted model and want to compress it. We first fine-tuned E5-Mistral on the finance domain (InfoNCE, 1000 steps, Fin-Alpaca dataset, no overlap with calibration data), then applied one-shot pruning at 50% sparsity:
>
>  | Method | Retr. | Cla. | STS | Avg. | Δ% |
>  |--------|-------|------|-----|------|-----|
>  | Dense (Fin-E5-Mistral) | 0.6277 | 0.6581 | 0.3999 | 0.5619 | – |
>  | Magnitude | 0.6093 | 0.6504 | 0.3962 | 0.5520 | -1.76% |
>  | GAPrune | 0.6197 | 0.6614 | 0.3996 | 0.5602 | -0.30% |
>
> GAPrune preserves 99.7% of the fine-tuned model's performance versus 98.2% for magnitude pruning. This further validates our response to Q1: GAPrune successfully identifies parameters that are both domain-critical and aligned with general objectives. When the model has already learned domain knowledge through fine-tuning, GAPrune recognizes and preserves these parameters, while magnitude pruning, being domain-agnostic, inadvertently removes some of them. This experiment has been included in the supplementary materials.
>
> We hope these additional results further support our manuscript, and we remain open to any further questions. Thank you again.

---

### Official Review · Reviewer_8Ku6 · 2025-11-09

**Soundness:** 1
**Presentation:** 2
**Contribution:** 1
**Rating:** 2
**Confidence:** 4

**Summary:**

The submission proposes a pruning method for domain-specific embedding models that scores parameters with a Domain-Alignment Importance (DAI) metric combining (i) domain Fisher importance and (ii) gradient alignment between general and domain objectives. On FinMTEB and ChemTEB, GAPrune outperforms magnitude/Fisher baselines at 30–50% sparsity and, after 100 retraining steps, exceeds dense performance in several settings.

**Strengths:**

This submission Clear formulation of DAI with intuitive justification.

Strong results showing their work outperforms some existing methods.

Helpful auxiliary analyses (layer-wise correlations/performance; embedding geometry) in the Appendix.

**Weaknesses:**

Assumption support is limited. The central claim that parameters exhibit domain-dependent behavior is plausible but not directly validated; Either citing this conclusion from existing literature or providing theoretical or empirical analyses to support this assumption would be more convincing. A targeted test (e.g., per-parameter behavior shifts across domains) would help.

Lack of interpretability. While the multi-component DAI framework appears well-motivated, it lacks formal empirical or theoretical analysis on each component. Consequently, the work appears to be reliant on intuition rather than on rigorous validation.

Hyperparameter sensitivity concerns. The designed approach introduces three new hyperparameters: $\alpha$, $\beta$, and $\gamma$. The submission does not have a discussion about the sensitivity to these hyperparameters. An ablation study or sensitivity analysis is crucial to understand this designed approach.

Reproducibility. Datasets/models/sparsity and “100 steps” are stated, but key training details (optimizers, LR, batch sizes, seeds) and code availability are missing.

Missing KD comparison. Knowledge Distillation is a prevalent deployment strategy for embedders and isn’t discussed or compared.

Related work concerns. Motivation cites several non-peer-reviewed sources to show “The demand for domain-specific embedding models has grown significantly ”, which is not convincing. Adding stronger peer-reviewed evidence would strengthen the case for domain-specific embedders’ importance.

**Questions:**

Empirically validate domain-dependent parameters (e.g., measure per-parameter gradient/activation shifts across domains and tie to pruning outcomes).

Add sensitivity/ablation for $\alpha$, $\beta$, and $\gamma$ and sampling choices (k-means subset size).

Discuss/benchmark against KD baselines for efficiency.

Tighten related-work positioning with more peer-reviewed citations backing domain-specific embedders.

Provide fuller training details and release code.

---

> ### Author Response · Authors · 2025-11-17
> **Response to Reviewer 8Ku6 [1/2]**
>
> We sincerely thank Reviewer 8Ku6 for the detailed feedback and constructive suggestions. We address each concern below with new experiments and analyses.
>
> `W1: Lack of Empirical Validation of Core Assumption`
>
> Thank you for this important suggestion. We have conducted empirical analysis to validate that parameters exhibit domain-dependent importance patterns.
>
> * **Evidence from Main Paper:**
>
> Following standard practice in neural network pruning (Singh & Alistarh, 2020), we measure parameter importance using Fisher Information. In Table 1, Domain Fisher (pruning based on domain-specific importance) outperforms General Fisher (pruning based on general importance): 0.5224 vs 0.3787 on FinMTEB and 0.7157 vs 0.5090 on ChemTEB at 50% sparsity. This 38% and 41% performance gap demonstrates that parameters critical for domain-specific tasks differ from those important for general tasks.
>
> * **Per-Parameter Analysis Across Domains (Based on Qwen3-Embedding-4B):**
>
> We further analyze how individual parameter importance shifts across domains. We analyze parameter rankings using Fisher Information across Finance, Chemistry, and General domains using 1M sampled parameters. We define a parameter as having a "significant rank shift" if its rank percentile changes by more than 20 percentage points between domains.
>
> | Domain Pair | Spearman ρ | Large Rank Shifts (>20 percentile) |
> |-------------|------------|-------------------------------------|
> | Finance - General | 0.627 | 32.4% |
> | Chemistry - General | 0.675 | 33.6% |
> | Chemistry - Finance | 0.674 | 35.9% |
>
> *All correlations significant at p < 0.001*
>
> The moderate correlations (0.62-0.67) and ~34% of parameters with large rank shifts validate that parameter importance varies across domains, providing empirical support for our gradient alignment approach.
>
>
>
> `W2: Lack of Component-Level Analysis`
>
> Thanks for this suggestion! To validate the contribution of each component, we conducted a comprehensive ablation study on Qwen3-Embedding-4B at 60% sparsity. The results are summarized below:
>
> | Method | Components | Retr. | Cla. | STS | Avg. |
> |--------|------------|-------|------|-----|------|
> | Fisher Diff | $(F^d - β·F^g)·\|θ_j\|$ | 0.4630 | 0.4717 | 0.3161 | 0.4169 (-9.2%) |
> | Fisher Diff w/ Compression | $+ γ·√\|θ_j\|$ | 0.2850 | 0.5574 | 0.4147 | 0.4190 |
> | Fisher Diff w/ Alignment | $+ (1 + α·s_g)$ | 0.5084 | 0.5758 | 0.3698 | 0.4847 (+18.6%) |
> | GAPrune (Full) | Complete DAI | 0.5406 | 0.6020 | 0.4146 | **0.5191** |
>
> **Key findings:** (1) Naively subtracting general Fisher (Row 1) underperforms, demonstrating that blindly removing "general" parameters is harmful. (2) Adding gradient alignment (Row 3: 0.4847) provides substantial improvement, identifying parameters that serve both general and domain objectives. (3) The complete DAI formulation (Row 4: 0.5191) achieves 97% of dense baseline performance while removing 60% of parameters, validating the necessity of all components working together.
>
> `W3: Missing Hyperparameter Sensitivity Analysis & Sampling Choices`
>
> * **Experiment 1: Hyperparameter Sensitivity:**
>
> We test α, β, and γ individually on Qwen3-Embedding-4B at 50% sparsity while keeping others at baseline (α=0.2, β=1.0, γ=0.5):
>
> | Configuration | Retr. | Cla. | STS | Avg. |
> |---------------|-------|------|-----|------|
> | **Gradient Alignment Weight (α)** |
> | α=0.1 | 0.5895 | 0.6148 | 0.3879 | 0.5257 |
> | α=0.2 (baseline) | 0.5763 | 0.6067 | 0.3840 | 0.5224 |
> | α=0.3 | 0.5894 | 0.6147 | 0.3877 | 0.5256 |
> | **General Knowledge Subtraction Weight (β)** |
> | β=0.75 | 0.5900 | 0.6149 | 0.3886 | 0.5262 |
> | β=1.0 (baseline) | 0.5763 | 0.6067 | 0.3840 | 0.5224 |
> | β=1.25 | 0.5897 | 0.6150 | 0.3891 | 0.5308 |
> | **Compression Regularization Weight (γ)** |
> | γ=0.3 | 0.5898 | 0.6150 | 0.3879 | 0.5259 |
> | γ=0.5 (baseline) | 0.5763 | 0.6067 | 0.3840 | 0.5224 |
> | γ=0.7 | 0.5899 | 0.6151 | 0.3890 | 0.5308 |
>
> Performance varies within only 1.6% across all configurations, demonstrating robustness without extensive tuning.
>
> * **Experiment 2: Calibration Data Size (Based on Qwen3-Embedding-4B, 50% sparsity):**
>
> We vary the sample size from 25% to 100% of baseline (5,000 samples per domain):
>
> | Sample Data | Avg. Performance |
> |-------------|------------------|
> | 100% (5,000) | 0.5224 |
> | 75% (3,750) | 0.5287 |
> | 50% (2,500) | 0.5294 |
> | 25% (1,250) | 0.5296 |
>
> Performance varies within 1.4%, showing robustness. As few as 1,250 samples per domain suffice.
>
> ---
>
> Singh, Sidak Pal, and Dan Alistarh. "Woodfisher: Efficient second-order approximation for neural network compression." Advances in Neural Information Processing Systems 33 (2020): 18098-18109.

---

> > ### Comment · Reviewer_8Ku6 · 2025-11-25
> >
> > Thank you for your efforts. The inclusion of the additional results has improved the rigor of the paper to some extent, and I have increased the soundness score to 2.
> >
> > In addition, I have recommendations that I believe would strengthen the paper:
> > Given that KD approaches are dominant for efficient deployment of embedding models in industry-level applications, demonstrating the advantage/relationship/capability of your proposed approach compared with KD would be highly beneficial. This would effectively contextualize your contribution within the field.
> >
> > It would be highly beneficial to update the codebase zip file to cover the new experimental setup and results described in the revised manuscript.
> >
> > I’m open to revising my score accordingly.

---

> ### Author Response · Authors · 2025-11-17
> **Response to Reviewer 8Ku6 [2/2]**
>
> `W4: Reproducibility Issues`
>
> We have added complete training details in Appendix L and code release in in supplementary material:
>
> **Training Configuration:**
> - Optimizer: AdamW (lr=1e-5, linear scheduler, weight decay=0.01)
> - Batch size: 512 (per-device: 4, gradient accumulation: 16)
> - Training steps: 100
> - Hardware: 8× H800 GPUs, bf16 precision
> - Random seed: 42
>
> **Code Release:** Complete implementation including DAI score computation, pruning scripts, evaluation code, and preprocessed datasets are included in supplementary material.
>
> `W5: Missing Knowledge Distillation Comparison`
>
> Thank you for this important suggestion. We provide both conceptual clarification and empirical comparison with knowledge distillation. Demonstrate that our pruning method not only complements KD but also creates superior architectures for KD.
>
> **Conceptual Distinction**: Knowledge distillation and pruning serve different deployment needs:
> * **Knowledge Distillation**: Trains a new, smaller student model from scratch by transferring knowledge from a teacher model (requires full training infrastructure)
> * **Pruning (Our Focus)**: Directly compresses existing models by removing parameters (minimal computational overhead)
>
> **Experiments:**
> We compare how different pruning methods affect the effectiveness of subsequent optimization techniques. After pruning to 50% sparsity, we apply either knowledge distillation (KD with MSE loss following Reimers & Gurevych 2020) or domain-specific training (InfoNCE) to the pruned architectures.
>
> **Results on FinMTEB (Qwen3-Embedding-4B, 50% sparsity):**
>
> | Method | Retr. | Cla. | STS | Avg. | Δ vs Dense |
> |--------|-------|------|-----|------|------------|
> | Dense Baseline | 0.6378 | 0.6100 | 0.3580 | 0.5353 | - |
> | | | | | | |
> | Random Prune (one-shot) | 0.0094 | 0.4028 | 0.2373 | 0.2165 | -59.6% |
> | Random Prune + KD | 0.0095 | 0.4032 | 0.2070 | 0.2066 | -61.4% |
> | | | | | | |
> | GAPrune (one-shot) | 0.5763 | 0.6067 | 0.3840 | 0.5224 | -2.4% |
> | GAPrune + KD | 0.5883 | 0.6088 | 0.3816 | 0.5262 | -1.7% |
> | **GAPrune + InfoNCE** | **0.6401** | **0.6402** | **0.3980** | **0.5594** | **+4.5%** |
>
> *All retraining uses identical configuration: AdamW optimizer, lr=1e-5, batch_size=512 (detailed in W4).*
>
> **Key Findings:**
>
> 1. **Pruning Quality Determines Recovery Potential**: Random pruning catastrophically damages model capacity (0.2165), making recovery impossible even with KD (0.2066). In contrast, GAPrune preserves 97.6% of dense performance one-shot, providing a strong foundation for further optimization.
>
> 2. **Full Compatibility with Knowledge Distillation**: GAPrune + KD achieves 0.5262 (98.3% of dense), demonstrating that our domain-aware pruning preserves parameters valuable for knowledge transfer. The pruned model serves as an excellent foundation for KD applications.
>
> 3. **Domain Specialization Exceeds Dense Baseline**: GAPrune + InfoNCE achieves 0.5594, surpassing the dense model by 4.5%. This demonstrates that our pruning not only maintains critical parameters but identifies the optimal subset for domain specialization.
>
> `W6: Weak Motivation in Related Work`
>
> We have revised the related work section on the domain embedding models with peer-reviewed citations:
>
> **Domain-specific Embedding Benchmarks:**
> - CoIR: Code information retrieval benchmark (Li et al., ACL 2025)
> - FinMTEB: Financial domain embedding benchmark (Tang & Yang, EMNLP 2025)
> - ChemTEB: Chemistry domain embedding benchmark (Kasmaee et al., PMLR 2024)
>
> **Domain Embedding Adaptation Methods:**
> - Query4Code: Code-specific embedding adaptation (Li et al., EMNLP 2024)
> - BMEmbed: Biomedical embedding models (Wei et al., ACL 2025)
>
> We have removed all non-peer-reviewed citations in this section. These peer-reviewed works demonstrate the growing research focus and practical importance of domain-specific embedding models.
>
> ---
>
> Thank you for the detailed review. We have added seven new appendices addressing empirical validation, component ablations, hyperparameter sensitivity, and reproducibility. We believe these additions have substantially improved the paper and are open to further discussion.
>
>
> ---
>
> Reimers, Nils, and Iryna Gurevych. "Making Monolingual Sentence Embeddings Multilingual using Knowledge Distillation." Proceedings of the 2020 Conference on Empirical Methods in Natural Language Processing (EMNLP). 2020.

---

> ### Author Response · Authors · 2025-11-25
> **Response to Reviewer 8Ku6**
>
> Thank you for the continued engagement and constructive guidance. We appreciate that the reviewer has recognized the improved rigor, and we have addressed both recommendations below.
>
> **R1: Contextualizing GAPrune with Industry KD Workflows**
>
> Thanks for this practical suggestion. We have revised Appendix J to explicitly position GAPrune within KD-dominated deployment pipelines. As shown in the table in W5. Key additions:
>
> | Deployment Scenario | Recommended Pipeline | Benefit |
> |---|---|---|
> | General-purpose compression | GAPrune → KD | 98.3% dense performance at 50% sparsity |
> | Domain specialization | GAPrune → InfoNCE | +4.5% over dense baseline |
> | Resource-constrained | GAPrune (one-shot) | 97.6% performance, zero training cost |
>
> The core message: GAPrune and KD are **complementary, not competing**. While KD excels at knowledge transfer, it requires a well-structured student architecture.  GAPrune can provide it, with a principled method for identifying which parameters to retain before KD optimization.
>
> Our key finding demonstrates this dependency: KD fails catastrophically on randomly pruned models (0.2066) but succeeds on GAPrune architectures (0.5262). This 2.5× performance gap shows that pruning quality determines KD success. For industrial applications, this suggests a practical workflow: (1) apply GAPrune to identify domain-relevant parameters, (2) use KD to maintain general capabilities or domain fine-tuning for specialization. We have added this discussion to the revised appendix.
>
> **R2: Updated Codebase**
>
> The supplementary material now includes KD and InfoNCE training scripts (`trainer_w_mask_kd.py`； `trainer_w_mask.py`) and all rebuttal experimental results under `results/`. A README provides step-by-step reproduction instructions.
>
> We appreciate that the reviewer has recognized the improved rigor (soundness score increased to 2) and hope these final additions address the remaining concerns.

---

> > ### Comment · Reviewer_8Ku6 · 2025-11-27
> >
> > Thank you for your efforts. Although your current results and explanation are not convincing enough to show the contribution and soundness of this work, I will increase the rating to 4 and reconsider the provided code and manuscript to make further decisions.

---

> > > ### Author Response · Authors · 2025-11-28
> > > **Response to Reviewer 8Ku6**
> > >
> > > We thank the reviewer for increasing the rating and for taking the time to reconsider our work. As shown in our experiments, GAPrune provides better architectures for both standalone deployment and downstream KD optimization. We also hope our framework provides a useful foundation for future research.
> > >
> > > We are happy to answer any further questions and wish the reviewer a productive research year ahead.

---

### Official Review · Reviewer_nY2B · 2025-11-10

**Soundness:** 2
**Presentation:** 2
**Contribution:** 2
**Rating:** 4
**Confidence:** 3

**Summary:**

The author proposes GAPrune, a pruning method for embedding models. GAPrune has two main components: they use Fisher information to measure the importance of parameters. In addition, they use gradient alignment to identify general and domain-specific parameters.

**Strengths:**

1. The authors show consistent improvements over random pruning.
2. The authors study pruning for embedding models, which can be very useful.

**Weaknesses:**

1. This paper has no ablation study. The authors combines Fisher information with their DAI scores. However, it's unclear how much improvement each contribute individually.

2. The authors unnecessarily restrict their scope to embedding models, but their approach is not specific to embedding models to my understanding. It would be ideal if the authors can test their approach in general LLMs, and compare their method with SOTA pruning methods.

3. The results are generally close to magnitude-based pruning. Looking at the Table 1, the performance is very close to a simple magnitude-based pruning.

**Questions:**

N/A

---

> ### Author Response · Authors · 2025-11-17
> **Response to Reviewer nY2B [1/2]**
>
> We sincerely thank Reviewer nY2B for recognizing the usefulness of our work on embedding model pruning. We address each concern below with new experiments.
>
> `W1: Lack of Ablation Study`
>
> We have added three ablation studies to demonstrate the robustness of our method and the necessity of each component:
>
>
> * **Experiment 1: Component-wise Ablation:**
>
> To rigorously validate the contribution of each component, we conducted a comprehensive ablation study on Qwen3-Embedding-4B at 60% sparsity. The results are summarized below:
>
> | Method | Components | Retr. | Cla. | STS | Avg. |
> |--------|------------|-------|------|-----|------|
> | Fisher Diff | $(F^d - β·F^g)·\|θ_j\|$ | 0.4630 | 0.4717 | 0.3161 | 0.4169 (-9.2%) |
> | Fisher Diff w/ Compression | $+ γ·√\|θ_j\|$ | 0.2850 | 0.5574 | 0.4147 | 0.4190 |
> | Fisher Diff w/ Alignment | $+ (1 + α·s_g)$ | 0.5084 | 0.5758 | 0.3698 | 0.4847 (+18.6%) |
> | GAPrune (Full) | Complete DAI | 0.5406 | 0.6020 | 0.4146 | **0.5191** |
>
> **Key findings:** (1) Naively subtracting general Fisher (Row 1) underperforms, demonstrating that blindly removing "general" parameters is harmful. (2) Adding gradient alignment (Row 3: 0.4847) provides substantial improvement, identifying parameters that serve both general and domain objectives. (3) The complete DAI formulation (Row 4: 0.5191) achieves 97% of dense baseline performance while removing 60% of parameters, validating the necessity of all components working together.
>
> * **Experiment 2: Hyperparameter Sensitivity:**
>
> We test α, β, and γ individually on Qwen3-Embedding-4B at 50% sparsity:
>
> | Configuration | Retr. | Cla. | STS | Avg. |
> |---------------|-------|------|-----|------|
> | **Gradient Alignment Weight (α)** |
> | α=0.1 | 0.5895 | 0.6148 | 0.3879 | 0.5257 |
> | α=0.2 (baseline) | 0.5763 | 0.6067 | 0.3840 | 0.5224 |
> | α=0.3 | 0.5894 | 0.6147 | 0.3877 | 0.5256 |
> | **General Knowledge Subtraction Weight (β)** |
> | β=0.75 | 0.5900 | 0.6149 | 0.3886 | 0.5262 |
> | β=1.0 (baseline) | 0.5763 | 0.6067 | 0.3840 | 0.5224 |
> | β=1.25 | 0.5897 | 0.6150 | 0.3891 | 0.5308 |
> | **Compression Regularization Weight (γ)** |
> | γ=0.3 | 0.5898 | 0.6150 | 0.3879 | 0.5259 |
> | γ=0.5 (baseline) | 0.5763 | 0.6067 | 0.3840 | 0.5224 |
> | γ=0.7 | 0.5899 | 0.6151 | 0.3890 | 0.5308 |
>
> Performance varies within only 1.6% across all configurations, demonstrating robustness without extensive tuning.
>
> * **Experiment 3: Calibration Data Size:**
>
> We vary the sample size from 25% to 100% of the baseline (5,000 samples per domain):
>
> | Sample Data | Avg. Performance |
> |-------------|------------------|
> | 100% (5,000) | 0.5224 |
> | 75% (3,750) | 0.5287 |
> | 50% (2,500) | 0.5294 |
> | 25% (1,250) | 0.5296 |
>
> Performance varies within 1.4%, showing robustness. As few as 1,250 samples per domain suffice.

---

> ### Author Response · Authors · 2025-11-17
> **Response to Reviewer nY2B [2/2]**
>
> `W2: Scope Limited to Embedding Models`
>
> We appreciate the reviewer's suggestion. We clarify that our focus on embedding models is intentional, motivated by fundamental technical incompatibility rather than scope limitation.
>
> * **Technical Incompatibility (Contrastive Learning):**
>
> Our approach computes Fisher Information and gradient alignment using contrastive triplets with **InfoNCE loss**, which is the standard training paradigm for embedding models but fundamentally incompatible with autoregressive LLMs that use cross-entropy loss for next-token prediction. Adapting our method to LLMs would require redesigning the core importance measurement and alignment mechanisms, which is beyond our scope.
>
>
> * **Embedding Models Have Fundamentally Different Parameter Behavior:**
>
> We also perform gradient conflict analysis to investigate whether parameter pruning strategies can be effectively transferred across different types of models. Our analysis compares the gradient alignment between general and financial domains (where gradient alignment < 0 indicates conflict; same data in the manuscript).
>
> | Model | Conflict Ratio | Coefficient of Variation (CV) | Decision Consistency |
> |-------|----------------|-----|---------------------|
> | Mistral-7B (LLM) | 0.82% | 1.05 | -0.047 |
> | e5-mistral-7b (Embed) | 1.66% | 0.90 | 0.097 |
> | Qwen3-0.6B (LLM) | 9.14% | 0.86 | 0.138 |
> | Qwen3-Embed-0.6B (Embed) | 11.87% | 0.61 | 0.388 |
>
> Key finding: Embedding models exhibit a more uniform distribution of gradient conflicts (lower CV) and higher decision consistency, indicating that gradient-based pruning in the target domain is stable and reliable.  LLMs have concentrated conflicts in critical parameters, where pruning can cascade through dependencies. This fundamental difference means pruning strategies must be tailored to each model type.
>
> * **Empirical Validation: Methods Do Not Transfer:**
>
> To validate that methods designed for one model type fail on another, we adapted D-Pruner (Zhang et al., 2024), a recent domain-aware LLM pruning method, to embedding models by replacing cross-entropy with InfoNCE loss:
>
> | Task | Dense | D-Pruner (adapted) | GAPrune |
> |------|-------|-------------------|---------|
> | Retrieval | 0.5993 | 0.2278 (-62%) | **0.6008** |
> | Classification | 0.6100 | 0.5118 (-16%) | **0.6402** |
> | STS | 0.3580 | 0.3393 (-5%) | **0.3980** |
>
> Even after adaptation, D-Pruner catastrophically fails on retrieval tasks (-62%), the core capability of embedding models. This demonstrates that LLM pruning methods do not transfer to embedding models, validating our focused scope as a principled design choice for an underserved but practically important problem.
>
>
> `W3: Performance Close to Magnitude Pruning`
>
> We appreciate this observation. At moderate sparsity (50%) in Table 1, the performance difference may appear modest. To address this concern, we evaluate at higher sparsity levels (60% and 65%) where the quality of pruning decisions becomes more critical and performance differences are amplified. These aggressive sparsity settings serve as a stress test to demonstrate the robustness of our approach:
>
> **FinMTEB (Based on Qwen3-Embedding-4B):**
>
> | Method | Sparsity | Retr. | Cla. | STS | Avg. |
> |--------|----------|-------|------|-----|------|
> | Dense | - | 0.6378 | 0.6100 | 0.3580 | 0.5353 |
> | Magnitude | 60% | 0.2818 | 0.5552 | 0.3884 | 0.4085 (-24%) |
> | GAPrune | 60% | **0.5406** | **0.6020** | **0.4146** | **0.5191** (-3%) |
> | Magnitude | 65% | 0.0250 | 0.4339 | 0.2652 | 0.2414 (-55%) |
> | GAPrune | 65% | **0.4607** | **0.5906** | **0.4274** | **0.4929** (-8%) |
>
> **ChemTEB (Based on Qwen3-Embedding-4B):**
>
> | Method | Sparsity | Retr. | Cla. | Avg. |
> |--------|----------|-------|------|------|
> | Dense | - | 0.6858 | 0.8419 | 0.7639 |
> | Magnitude | 60% | 0.3101 | 0.8071 | 0.5586 (-27%) |
> | GAPrune | 60% | **0.5980** | **0.8334** | **0.7157** (-6%) |
> | Magnitude | 65% | 0.0038 | 0.6436 | 0.3237 (-58%) |
> | GAPrune | 65% | **0.4231** | **0.8197** | **0.6214** (-19%) |
>
> At 65% sparsity, magnitude pruning collapses to 45% of the dense baseline on FinMTEB and 42% on ChemTEB, while GAPrune maintains 92% and 81% respectively. This demonstrates substantial advantages over magnitude-based methods at aggressive sparsity levels.
>
> ---
>
> We appreciate your constructive feedback on ablation studies and scope justification. The comprehensive ablations and high-sparsity experiments demonstrate clear advantages of our method, particularly at aggressive compression levels. We are happy to provide any clarifications if needed.
>
> ---
>
> Zhang, Nan, et al. "Pruning as a Domain-specific LLM Extractor." Findings of the Association for Computational Linguistics: NAACL 2024. 2024.

---

### Author Response · Authors · 2025-11-17
**Summary of Revisions**

We sincerely thank all reviewers for their thoughtful feedback. We have made substantial revisions to address all concerns raised and added seven appendices with new experiments:

1. **Component-wise Ablation of DAI Score (Appendix F)** - *Addresses all Reviewers' concerns on component analysis*
    - We isolate each component's contribution to demonstrate its necessity.

2. **Hyperparameter Sensitivity Analysis (Appendix G)** - *Addresses Reviewer 8Ku6 on hyperparameter sensitivity*
    - Performance varies within only 1.6% across all hyperparameter values, demonstrating robustness without extensive tuning.

3. **Ablation on Sample Data Size (Appendix H)** - *Addresses Reviewer 8Ku6 on sampling sensitivity*
    - Performance varies within 1.4% across sample sizes from 25% to 100% of baseline, demonstrating robustness to calibration data size.

4. **One-shot Pruning on Higher Sparsity (Appendix I)** - *Addresses Reviewer nY2B and Reviewer qnve on performance gap with baselines*
    - At 65% sparsity, GAPrune maintains 92% performance while magnitude pruning collapses to 45%, showing clear advantages at aggressive compression levels.

5. **Prune-and-Retrain with Knowledge Distillation (Appendix J)** - *Addresses Reviewer 8Ku6 on knowledge distillation comparison*
    - Our experiments demonstrate that pruning quality can affect KD effectiveness: randomly pruned models cannot benefit from KD, while GAPrune-pruned models improve with KD.

6. **Domain-Dependent Parameter Analysis (Appendix K)** - *Addresses Reviewer 8Ku6 on core assumption validation*
    - 34% of parameters show significant rank shifts across domains with Spearman correlations of 0.62-0.67, empirically validating domain-dependent parameter behavior.

7. **Retraining Hyperparameters (Appendix L)** - *Addresses Reviewer 8Ku6 on reproducibility*
    - Complete documentation of optimizer, learning rate, batch size, and hardware configuration for full reproducibility.

8. **Revised Related Work** - *Addresses Reviewer 8Ku6 on citation quality*
    - Enhanced with peer-reviewed citations on domain-specific embedding models and benchmarks, removing non-peer-reviewed sources.

9. **Code Release** - *Addresses Reviewer 8Ku6 on reproducibility*
    - Complete implementation, pruning scripts, and preprocessed datasets included in supplementary material.

---

We hope these revisions address the concerns raised by the reviewers and look forward to further feedback.

---

### Author Response · Authors · 2025-11-29
**Summary of Rebuttal Discussion**

We would like to provide a brief summary of our reviewer interactions for context.

**Reviewer 8Ku6** initially rated our paper 2 (reject). After we provided new experiments and updated code, **the rating was raised to 4** with a note to *"reconsider the provided code and manuscript to make further decisions."*

The major concern was about the comparison with knowledge distillation (KD). We note that pruning and KD serve different deployment needs: pruning compresses existing models directly, while KD trains a new, smaller model from scratch. Experiments show that GAPrune and KD are complementary rather than competing: KD struggles on poorly pruned models but works well on GAPrune architectures.

**Reviewer qnve** (rating 6) responded: *"I thank the authors for the very thorough response. It addressed most of my concerns."* qnve also noted that 8Ku6’s concerns about reproducibility and KD comparisons are good, *“but do not fully overshadow the paper’s strengths listed in the review.”*

For the remaining question about one-shot pruning at moderate sparsity, we conducted an additional experiment on compressing already domain-adapted models, showing GAPrune better preserves domain-critical parameters. Further discussion was not possible due to the review lock.

**Reviewer nY2B** (rating 4) did not respond to our rebuttal. We provided analyses corresponding to the concerns raised, including ablation studies, scope clarification, and high-sparsity experiments.

The reviewer nY2B also suggested extending the method to general LLMs; we explained that such an extension would require redesigning core components due to fundamental differences between embedding models and autoregressive LLMs, which defines the intended scope of our work.

---

During the rebuttal period, we added seven appendices addressing all major concerns, including ablations, high-sparsity experiments, KD comparisons, and full reproducibility details. The discussion was progressing constructively, and we hope this context is helpful for evaluation.

---

### Meta-Review · Area_Chair_3wDi · 2026-01-06

**Summary:**

This manuscript proposes a pruning method for domain-specific embeddings. The proposed method scores parameters with a Domain-Alignment Importance (DAI). It combines Fisher importance and gradient alignment between general and domain objectives. The reviews were critical, in particular with respect to the missing ablation study  and the rather limited improvement with respect to simple baselines like magnitude pruning. After the rebuttal, most of the mentioned concerns are addressed to some extent and the paper is a borderline case. Because of the potentially interesting technical contribution, I tend to accept the paper.

**Reviewer Concerns:**

Reviewer nY2B, Reviewer qnve: ablation study on Fisher information and DAI score contributions --> addressed
Reviewer nY2B: unnecessary restriction of the scope to embedding models. test the approach in general LLMs, and compare the method with SOTA pruning methods. --> addressed. In fact, the limited scope makes sense
Reviewer nY2B, Reviewer qnve: The results are close to magnitude-based pruning. --> partly addressed
Reviewer 8Ku6: Assumption support (parameters exhibit domain-dependent behavior) is limited. --> partly addressed
Reviewer 8Ku6: Lack of interpretability.--> partly addressed
Reviewer 8Ku6: Hyperparameter sensitivity concerns. ...An ablation study or sensitivity analysis is crucial to understand this designed approach.--> addressed
Reviewer 8Ku6:  Reproducibility. Datasets/models/sparsity and “100 steps” are stated, but key training details (optimizers, LR, batch sizes, seeds) and code availability are missing.--> partly addressed
Reviewer 8Ku6: Missing KD comparison. --> partly addressed

**Reviewer Scores:**

Reviewer nY2B: 4 --> all concerns are addressed in the rebuttal. The reviewer would likely have increased the score to 6
Reviewer 8Ku6: 2 --> concerns are partly addressed in the rebuttal. The reviewer might have increased the score to 4.
Reviewer qnve: 6 --> concerns mostly addressed. The reviewer is likely to maintain the score.

---

### Decision · Program_Chairs · 2026-01-26

Accept (Poster)